# Benchmarking Uncertainty Estimation in Large Language Model Replies for Natural Science Question Answering

## Abstract

Large Language Models (LLMs) are commonly used in Question Answering (QA) settings, increasingly in the natural sciences if not science at large. Reliable Uncertainty Quantification (UQ) is critical for the trustworthy uptake of generated answers. Existing UQ approaches remain weakly validated in scientific QA, a domain relying on fact-retrieval and reasoning capabilities. We introduce the first large-scale benchmark for evaluating UQ metrics in scientific QA studying calibration of UQ methods, providing an extensible open-source framework to reproducibly assess calibration. Our study spans up to 20 large language models of base, instruction-tuned and reasoning variants. Our analysis covers seven scientific QA datasets, including both multiple-choice and arithmetic question answering tasks. We evaluate and compare methods representative of prominent approaches on a total of $685,000$ long-form responses, spanning different reasoning complexities representative of domain-specific tasks. At the token level, we find that instruction tuning induces strong probability mass polarization, reducing the reliability of token-level confidences as estimates of uncertainty. Models further fine-tuned for reasoning are exposed to the same effect, but the reasoning process appears to mitigate it depending on the provider. At the sequence level, we show that verbalized approaches are systematically biased and poorly correlated with correctness, while answer frequency (consistency across samples) yields the most reliable calibration. In the wake of our analysis, we study and report the misleading effect of relying exclusively on ECE as a sole measure for judging performance of UQ methods on benchmark datasets. Our findings expose critical limitations of current UQ methods for LLMs and standard practices in benchmarking thereof.

## 1 Introduction

Large Language Models (LLMs) have rapidly emerged as powerful tools for natural language processing, understanding and generation. Among their diverse applications, they are increasingly deployed in chat-based assistants for Question Answering (QA), automated agent-based systems, and serving as surrogates for traditional search engines (Jin et al., 2025; Xiong et al., 2024a; Kelly et al., 2023). Within this landscape, scientific QA constitutes a particularly critical and challenging task. It spans a wide range of use cases, from public science communication (Schäfer, 2023) and education across different levels of expertise (Welbl et al., 2017), to supporting research and knowledge discovery (Gu & Krenn, 2025). In these contexts, accuracy and trustworthiness are essential, as errors may misinform the public, impair learning, or distort scientific practice. A central obstacle to reliability is the phenomenon of hallucinations, where LLMs generate fluent and seemingly confident answers that are factually incorrect or misleading (Ji et al., 2022). While hallucinations are now widely recognized, effective methods to detect and mitigate them remain underdeveloped, particularly in high-stakes domains such as science. Evidence for this can be found in UQ methods being absent from popular products while being subject to research since several years.

Research into UQ aims to develop reliable, automated methods for quantifying how certain a model is in its own predictions (Guo et al., 2017). In the context of LLMs, UQ serves as a critical tool to mitigate hallucinations: by identifying outputs with high uncertainty, systems can flag poten-

tially erroneous responses, abstain from answering, or route queries to alternative mechanisms – i.e. larger models, retrieval-augmented systems, or human reviewers. Beyond error mitigation, reliable UQ enhances transparency and user trustworthiness by providing interpretable indicators of answer reliability and soundness (Dhuliawala et al., 2023; Devic et al., 2025; Reyes et al., 2025).

## 2 KEY CONTRIBUTIONS

To address open questions regarding the applicability of UQ in scientific QA, this work provides a comprehensive benchmark of uncertainty estimation methods with a qualitative and quantitative focus on calibration. We identify and evaluate a set of core uncertainty estimation approaches that are widely used and form the basis of many derivative methods, covering three principal approaches: token-level confidences, verbalized approaches and semantic consistency.

We first select UQ methods based on applicability, computational feasibility, and relevance to current practice. We identify core challenges for LLMs and subsequent UQ on responses in the scientific domain. We evaluate the selected methods on a diverse collection of models and datasets, incorporating different difficulty levels we identified to make scientific QA especially challenging. For verifyable ground truth we are using multiple choice QA and arithmetic QA.

In total, we analyze 685,000 responses across datasets and models, making this one of the largest and most realistic evaluations of uncertainty quantification in long-form scientific QA to date. We report detailed summary statistics, distributional analyses, and probability plots, addressing limitations of prior evaluations that were predominantly conducted on short, saturated datasets and often only in the context of newly proposed methods.

By evaluating how the selected UQ methods capture the reliability of reasoning-heavy, long-form predictions, we characterize systematic model behaviour and fundamental limitations in reasoning-demanding settings where reliable uncertainty estimates are particularly valuable. By identifying systematic differences across settings, we uncover potential drivers of divergent UQ behaviour and point to mechanisms that warrant further investigation. These insights motivate future work to understand and mitigate the factors shaping uncertainty estimates in complex reasoning tasks.

We empirically evaluate the selected methods and analyze the resulting data to address two guiding research questions:

**(RQ1)** To what extent are the token probabilities a calibrated measure of confidence and what effect does the instruction-tuning or reasoning process have on the calibration?

**(RQ2)** To what extent do sequence-level uncertainty-quantification methods provide reliable uncertainty estimates for long-form answers in scientific question answering, across tasks spanning retrieval-based to reasoning-intensive questions?

Finally, to facilitate ongoing research and rapid iteration as LLMs and UQ methods continue to evolve, we introduce a flexible and extensible framework for LLM benchmarking. We release this framework together with the implementation of benchmarks for calibration assessment presented in this paper, detailed reproduction instructions, raw uncertainty scores from our benchmark runs, and additional visualizations in an open-access repository accompanying this paper[1].

## 3 BACKGROUND

### 3.1 UNCERTAINTY QUANTIFICATION IN PREDICTIVE MODELS

Neural networks, including LLMs, face predictive uncertainty as their training data provide only a discrete and incomplete mapping of real-world artifacts (Hüllermeier & Waegeman, 2021).The paradigm of negative log likelihood training for next token prediction (Radford et al., 2018) in LLM pre-training enforces uncertainty of generated tokens. Besides obtaining this predictive uncertainty with dedicated methods (Gal et al., 2016), the act of validating the quantitative soundness of uncertainties comes as a challenge to many practitioners (Guo et al., 2017; Chung et al., 2021).

---

[1]https://anonymous.4open.science/r/llm-uncertainty-bench-9B2B/

The field of UQ methods in LLMs is still nascent compared to methods for classification or regression tasks (Kendall & Gal, 2017; Kuleshov et al., 2018; Papamarkou et al., 2024) in general. In the broader UQ literature, two core tasks are typically distinguished: calibration, which evaluates the alignment between confidence scores and predictive accuracy, and selective prediction, which concerns deciding when a model should abstain(Liu et al., 2025a). In this work we focus on calibration, as it forms the statistical backdrop of almost all UQ methods and directly determines the reliability of predicted confidences. Since calibration evaluates predicted probabilities for correctness, only methods that produce scores in the $[0, 1]$ range are applicable. Arbitrary or unbounded scores are not directly usable for calibration metrics.

The standard evaluation technique in calibration are calibration plots, which visualise how well predicted confidence scores correlate to the true likelihood of correctness, and summary statistics thereof (Guo et al., 2017; Hendrycks & Gimpel, 2018). Many quantitative approaches rely on information-theoretic metrics, such as entropy or perplexity, but these often fail to capture language-specific nuances. Recently, LLM-specific uncertainty estimation methods have emerged, such as Verbalized Uncertainty, P(True), and Claim Conditioned Probability (CCP) (see details in Section 3.2 and Section 7.1). However, they are studied in narrow domains on simple tasks, such as factual QA on biographies or encyclopedic content (Fadeeva et al., 2024). Consequently, it remains unclear how well these approaches generalize to scientific QA, which poses the unique challenges outlined in Section 3.3. Compounding the issue, current benchmarks are often tightly coupled to specific models, datasets, and UQ methods, limiting adaptability to rapidly evolving LLM architectures and use cases.

## 3.2 MULTIPLE-CHOICE QUESTION ANSWERING AND UQ CALIBRATION

A straight forward way of estimating uncertainty in LLMs is to reformulate Multiple-Choice Question Answering (MCQA) items as classification tasks, prompting the model to output a single label token (e.g., A/B/C/D) representing the chosen answer. The confidence scores are derived from the probabilities assigned to each label. These probabilities can be used as confidence scores directly, i.e. ignoring probability mass on non-label tokens. Alternatively, a normalization with respect to the total probability mass assigned to all possible answer labels can be applied. By normalizing over the label set, the resulting confidence scores represent relative preferences among the labels – not (un-)certainty in the options. This may obscure uncertainty that would otherwise be expressed through low absolute probabilities, making the normalized scores less reliable as measures of uncertainty (Wang et al., 2024a). The GPT-4 Technical Report (OpenAI, 2023) compared the calibration of responses from a base model and instruction-tuned model using this approach. Their results suggested good calibration in the base model, but significantly worse calibration in the instruction-tuned model. This sparked a so far understudied controversy about the influence of instruction-tuning on the calibration of models which is often encountered in colloquial discussions.

## 3.3 CHALLENGES IN SCIENTIFIC QUESTION ANSWERING

Scientific QA presents challenges that extend far beyond factual recall of general MCQA. Questions often combine domain-specific terminology, symbolic expressions, and quantitative relationships, requiring models to interpret structured scientific information and integrate it into coherent reasoning processes. Many tasks demand the recall of appropriate formulae, the identification of relevant variables, and the manipulation of non-linear or multi-equation systems. These steps exceed simple pattern matching and due to the lack of intrinsic arithmetic ability, push LLMs toward brittle heuristics. Thus, scientific problem solving frequently involves complex, dependency-rich reasoning chains. Errors in intermediate steps propagate, rendering downstream inferences unreliable. This complicates UQ, as standard methods assume uncertainty is localized to individual predictions. Effective UQ in scientific QA requires accounting for cumulative uncertainty across multiple reasoning steps, each susceptible to distinct failure modes.

This dependency structure is precisely what makes scientific QA both difficult for current LLMs and uniquely important for evaluating UQ methods. Whereas short-form benchmarks hide these challenges, scientific QA exposes the complex, compounding nature of uncertainty in realistic, reasoning-intensive settings, where reliable UQ stands to provide the greatest practical benefit.

# 4 RELATED WORK

UQ is particularly critical for LLMs because their token-level probabilistic generation can produce fluent yet confidently incorrect or misleading outputs known as hallucinations (Maynez et al., 2020). Hallucinations, classified as intrinsic contradictions or extrinsic fabrications, diminish trustworthiness and require robust uncertainty detection to uncover them (Sui et al., 2024; Zhang et al., 2023b; Banerjee et al., 2024). Many methods have been established to obtain predictive uncertainties for LLM generated text (Geng et al., 2024).

Token-level uncertainty estimation faces challenges due to varying semantic importance of tokens. Many methods nevertheless assume equal token importance, leading to misrepresentations (Ullrich et al., 2025; Kuhn et al., 2023a). Linguistic calibration by epistemic markers (e.g., "might," "potentially") provides interpretable uncertainty cues, but models tend to be overconfident in these expressions, risking user over-reliance (Band et al., 2024; Zhou et al., 2024). Empirical work shows that base LLMs are generally better calibrated than instruction-tuned models, which often become overconfident. (OpenAI, 2023; Tian et al., 2023; Wang et al., 2025). While researching risk scores on a binary classification task in a narrow domain, (Cruz et al., 2024) reported a polarization of binary label probability scores as a consequence of instruction tuning. Whereas overconfidence refers to a systematic tendency for confidence scores to exceed actual accuracy, polarization denotes a collapse of the token-level probability distribution. In polarization, the model allocates nearly all probability mass to a single label. Given the constrained setting in (Cruz et al., 2024), the generalizability of this finding on token-level calibration remains unclear.

Prompt design significantly affects uncertainty and calibration: small prompt variations, the use of epistemic phrasing, and simulating knowledge profiles influence both accuracy and confidence. Strategies have been proposed to mitigate overconfidence in self-evaluations (Cao et al., 2024; He et al., 2024; Sclar et al., 2024; Zhou et al., 2023; Lu & Wang, 2024; Xiong et al., 2024b). This body of work underscores the complexity of uncertainty estimation in LLMs and the need for context-aware, robust and scalable analyses of UQ methods.

First comprehensive studies of UQ effectiveness as well as their calibration were undertaken only recently by Fadeeva et al. (2023a) and Huang et al. (2025). In Fadeeva et al. (2023a), the authors share open-source UQ benchmark tooling (`LM-Polygraph`) publicly, underlining the importance of UQ analyses as LLM architectures progress rapidly. This study focuses on selective prediction (Geng et al., 2024). Both studies rate the effectiveness of UQ methods by virtue of one (Fadeeva et al., 2023a) and two (Huang et al., 2025) summary statistics respectively. This stands in contrast to complementary best practices (calibration plots) of closely related ML fields to evaluate predictive uncertainties (Guo et al., 2017) both quantitatively and qualitatively. Multiple reports (Zhang et al., 2024a; Vashurin et al., 2025) around the `LM-Polygraph` project offer a structured comparison of UQ methods considering computational cost, logit access, and granularity of uncertainty of a wide range of uncertainty methods. But these reports target the task of selective prediction exclusively, enabling the use of methods producing unbound scores, and focus on one single summary statistic (Prediction Rejection Ratio, PRR) to compare UQ methods. Here, we identified a gap in existing academic literature which understudied the calibration of UQ methods in favor of selective prediction.

We do observe early studies of calibration as a measure of quality control for UQ in LLMs: Tao et al. (2025) presented a vast analysis with respect to explored models, but restrict themselves to one single dataset only. Liu et al. (2025b) surveyed a variety of published UQ methods and presented available datasets, but did not perform experiments for an empirical comparison.

Our work aims to complement these findings by introducing a benchmark that systematically evaluates methods on the second core task of UQ: calibration. We assess methods representative of the most prominent approaches on scientific QA, stress-testing them on long-form responses that require complex syntactic and multi-step reasoning. By incorporating a diverse range of models and datasets, our benchmark provides reproducible, transparent qualitative and quantitative evidence, offering a solid foundation for future research on calibration in LLMs.

## 5 General Experimental Setup

### 5.1 Dataset Selection

Due to the necessity of verifiable ground truth for computing calibration metrics such as ECE, AU-ROC and calibration plots, we target structured tasks. We use four multiple-choice QA datasets in combination with counterfactual prompting using the APriCoT prompting strategy to better approximate open-ended QA behaviour. To complement this, we are using three arithmetic QA datasets for better generalizability. Our evaluation includes datasets with a strong emphasis on scientific QA, containing content from physics, chemistry and biology. For low complexity arithmetic reasoning, math datasets are employed.

To study the effect of task complexity, the selected datasets span a range of difficulty levels and cognitive demands, from fact retrieval to syntactic and multi-step reasoning required in scientific QA. We treat scientific QA as a representative domain with high relevance for safety-critical and reasoning-intensive applications and with strong potential for generalisation beyond individual sub-fields. Through this selection, our benchmark is explicitly designed to evaluate uncertainty methods against the challenges identified in Section 3.1, ensuring that the tasks reflect the reasoning, domain knowledge and syntactic complexities that make scientific QA particularly demanding.

The following provides an overview over seleted datasets (more details provided in Section A.3).

**MMLU** (Hendrycks et al., 2021) is a multiple-choice benchmark with $15,908$ questions across $57$ academic subjects, including physics at varying levels. It is primarily testing comprehension, factual knowledge and single step reasoning.

**ARC** (Clark et al., 2018) tests scientific reasoning using grade-school science exam questions split into **ARC-Easy** and **ARC-Challenge** subsets. The latter emphasizes multi-step reasoning, making it a strong benchmark for evaluating UQ methods under complex conditions.

**SciQ** (Welbl et al., 2017) contains $13,679$ multiple-choice questions in physics, chemistry, and biology. It emphasizes conceptual understanding.

**GPQA** (Rein et al., 2023) is a graduate-level science benchmark with $448$ expert-written questions across physics, chemistry and biology, designed to resist simple lookup. Its high difficulty and reasoning demands make it particularly useful for stress-testing UQ methods.

**GSM8K** (Cobbe et al., 2021) is a standard math word problem dataset with $8,500$ arithmetic questions requiring step-by-step symbolic reasoning. **GSM-MC** (Zhang et al., 2024b) is a multiple-choice variant of GSM8K, using model-generated distractors to reduce evaluation ambiguity.

**SVAMP** (Patel et al., 2021) is an arithmetic reasoning dataset containing $1,000$ questions that introduce distracting information in the problem text. While featuring low to moderate computational complexity, these distractors are specifically designed to induce ambiguity and decision uncertainty, challenging models can recognize and quantify uncertainty in the presence of misleading or irrelevant information.

**SciBench** (Wang et al., 2024b) comprises 692 college-level questions from math, chemistry, and physics textbooks. It targets advanced symbolic reasoning involving formulas and physical units.

### 5.2 Model Selection

Models were chosen to cover a broad spectrum of LLMs while ensuring reproducibility through the use of open-weight, publicly available models. To capture diversity in design, the selection spans five major providers (OpenAI, Mistral, Meta, Qwen, and Google) and includes models of varying sizes (from 7B to 70B parameters), variants (base, instruction-tuned and reasoning models) and architectural designs, such as Mixture-of-Experts. To research the effect of instruction tuning on the calibration of label probabilities, instruction-tuned and reasoning models are complemented by their base model counterparts to enable controlled comparison. A detailed list of selected models can be found in Section A.2.

# 6 BENCHMARKING LABEL PROBABILITY CALIBRATION

## 6.1 METHODOLOGY / EXPERIMENTAL SETUP

While confidence scores represent a model's self-assessed probability of correctness, effective UQ requires calibration methods that align these scores with empirical accuracy (Guo et al., 2017). Expected Calibration Error (ECE) is a widely used metric for this purpose, comparing binned confidence estimates with actual correctness.

To ensure comparability while extending prior work, the experimental setup was designed to remain close to the GPT-4 Technical Report (OpenAI, 2023), while expanding in model coverage, dataset diversity and calibration and score distribution analysis. We evaluated all selected base, instruction-tuned and reasoning models across four MCQA datasets, chosen to represent different reasoning complexities: factual and single-step reasoning (MMLU), symbolic reasoning (GSM8K), and multi-step reasoning (ARC-Reasoning, GPQA). Initial tests revealed substantial differences in task comprehension between base and instruction-tuned models. To address this, we designed four alternative prompts (see Section A.6.1), each using three-shot prompting. We then selected the prompt that maximized the average probability mass assigned to label tokens, thereby ensuring task comprehension across models. For the final results, we employed structured decoding during generation to reliably obtain relative label probabilities and to prevent invalid answers, as detailed in Section A.6.4.

Calibration performance was then assessed by comparing the model variants using label probabilities as confidence scores with calibration plots and summary metrics such as ECE.

## 6.2 KEY FINDINGS

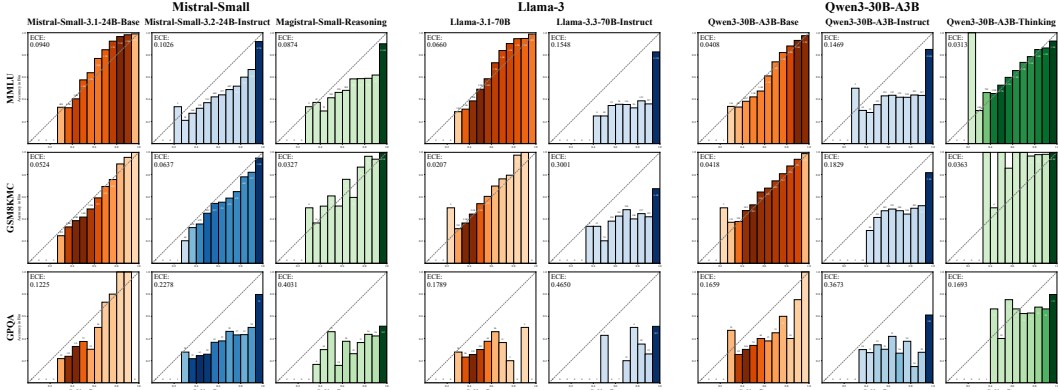

Figure 1: **Calibration Plots Using Label Probabilities as Uncertainty Scores for Only the Most Probable Label Per Prompt.** Columns correspond to three selected model families: base variants are shown in orange, instruction-tuned variants in blue, and reasoning variants in green. Rows refer to three QA datasets on MMLU, GSM8KMC and GPQA. Darker shading indicates a higher number of items within each confidence bin. Each bin is labelled with the sample count contained. The plot shows the results using Prompt Design 1 with structured decoding enabled.

Figure 1 reports a subset of representative calibration plots. All results shown use Prompt Design 1 (see Section A.6.1), which was found to optimize task comprehension across model types (see Section A.6.2). Structured decoding has been employed to generate the final result, after normalization has been validated (see Section A.7.2). Comprehensive results for all models, datasets and prompt designs are included in Section A.7.1.

We find that ECE increases with reasoning complexity (see Figure 1). Tasks requiring symbolic or multi-step reasoning, such as GSM8K and GPQA, exhibit substantially higher ECE compared to factual retrieval tasks, such as MMLU. This distinction highlights that token-level probabilities can reliably capture aleatoric uncertainty in fact retrieval, but become overconfident and unreliable when reasoning is required as a result of epistemic uncertainty.

We reproduce the degradation of ECE observed by (OpenAI, 2023) as a result of intruction-tuning, although not uniformly across models to the same degree. However, visual analysis of calibration plots reveals systematic polarization of token probability distributions (see Figure A.2). Instruction-tuned models tend to concentrate nearly all probability mass on a single label compared to their base model counterparts, thereby degrading the expressiveness of token-level confidence scores. This effect is evident in Figure 1, which shows a pronounced accumulation of items in the highest-confidence bucket. The degree of polarization is consistent across most instruction-tuned model families, with few notable exceptions among Mistral models.

This effect is also evident in the *gpt-oss* and *Magistral* reasoning models, where the reasoning process appears to commit to a single option. In contrast, the *DeepSeek-R1* and *Qwen3* reasoning model families seem to consider multiple options and actively retrieve relevant facts rather than committing prematurely. This allows the model's uncertainty estimates to better reflect uncertainty arising from the different possible answers, resulting in improved calibration and reduced polarization. Because these differences cluster by model provider rather than by scale or architecture, we hypothesize that provider-specific training pipelines are a key underlying factor. Our benchmark thus provides a foundation for further research into how individual steps in training influence model calibration and polarization.

## 6.3 STRUCTURAL LIMITATIONS OF ECE FOR CALIBRATION ASSESSMENT

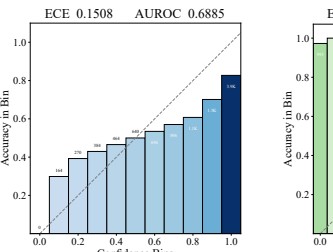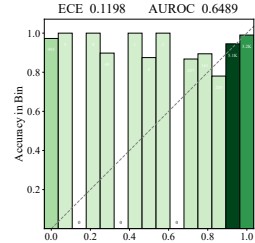

Figure 2: **ECE as a Misleading Proxy for Calibration: An Illustrative Case.** Two calibration plots from our results illustrate that similar or even lower ECE values do not necessarily indicate better calibration. In the left plot, uncertainty scores exhibit a clear relationship with correctness. In the right plot, there is no meaningful relationship between scores and correctness, yet the model still obtains a lower ECE and only a slightly lower AUROC.

The mixed levels of ECE degradation observed in the Label Probability Experiment, see Figure 1, highlights a fundamental limitation of ECE as a calibration metric. Although ECE measures the correlation between predicted confidence and the likelihood of correctness, it is not independent of the model's overall accuracy. As a result, when confidence scores collapse into a narrow region of the distribution, ECE becomes driven primarily by overall accuracy rather than by the reliability of individual uncertainty estimates. Consequently, as model accuracy increases and consistently high confidences are reported, ECE may appear deceptively low even when the model remains poorly calibrated at the instance level, see Figure 2. This undermines the utility of ECE for UQ, where the primary goal is to assess whether a method can accurately quantify uncertainty on a per-instance basis, independent of its overall accuracy. This limitation is particularly relevant for LLMs, due to our findings of confidence polarization, where models produce consistently very high confidences. This issue has serious implications for the evaluation of novel UQ methods and benchmarks, as prior studies that rely solely on summary metrics like ECE may report misleadingly optimistic results. We advocate for using ECE only in combination with visual inspection of calibration plots and complementary summary metrics.

## 7 BENCHMARKING SEQUENCE-LEVEL CALIBRATION

### 7.1 SELECTION OF UNCERTAINTY MEASURES

Sequence level methods often use token-level confidences (examined in Section 6) via different aggregation strategies. Beyond these methods, we identify two more approaches of sequence-level

UQ methods that meet our criteria of theoretical grounding, computational feasibility, and empirical promise: verbalized approaches and semantic consistency. We further consider their practical relevance and the extent to which they underpin recently proposed UQ methods. From these families, we select four prominent and representative methods for evaluation. A justification of our selection criteria, as well as the rationale for excluding other approaches is provided in Section A.5.

**Verbalized Uncertainty** (Tian et al., 2023) prompts the model subsequent to the answer generation to provide a self-assessed probability estimate of correctness in token space. While straightforward, it ignores token probability distributions and may be susceptible to training data bias.

**P(True)** (Kadavath et al., 2022) prompts the model subsequent to the answer generation to classify the answer as "(A) True" or "(B) False". The underlying token probabilities assigned to the corresponding labels "(A)" and "(B)" are then used as confidence scores.

**Frequency of Answer** estimates certainty by the proportion of semantically equivalent answers among multiple sampled generations for the same prompt. This is computationally expensive and semantic equivalence detection of answers is non-trivial in open ended questions answering. However, it captures semantic consistency and proxies other approaches like self-consistency prompting (Wang et al., 2023).

**Claim-Conditioned Probability (CCP)** (Fadeeva et al., 2024) evaluates uncertainty at the token level by determining the semantic consistency among the top probable token alternatives. This is done by clustering the token alternatives in tokens that entail and contradict the original meaning by comparing the chosen token with its alternatives using an NLI model. The token-level confidence score is the ratio of probability mass assigned to entailing tokens to the sum of entailing and contradicting tokens. Sequence-level confidence is calculated from product of token confidences.

## 7.2 METHODOLOGY / EXPERIMENTAL SETUP

To evaluate the calibration of different uncertainty methods in long-form QA, we focus exclusively on instruction-tuned and reasoning models, as base models exhibit poor task comprehension and are unsuitable for QA tasks. For this experiment, all seven previously selected MCQA and arithmetic QA datasets were used. MCQA introduces specific challenges: (1) **Selection bias.** Models may favor certain answer choices due to token frequency or formatting learned during training, regardless of semantic content (Myrzakhan et al., 2024). (2) **Positional bias.** Models can exhibit systematic preference for certain label positions (e.g., always selecting "A") (Zheng et al., 2024).

We address these biases using the APriCoT prompting strategy (Moore et al., 2025), which combines Chain-of-Thought reasoning with counterfactual prompting. Each answer choice is evaluated independently, and the model classifies it as correct or incorrect, producing a verifiable judgment. This isolates answer evaluation from ordering and formatting effects, reducing both selection and positional bias and improving calibration. Rephrasing MC questions as open-ended queries could avoid format-related biases, but many items depend on predefined choices (e.g., fill-in-the-blank or elimination logic). APriCoT also helps approximate open-ended QA by eliciting reasoning over individual answers while preserving a verifiable format.

For arithmetic QA datasets, Chain-of-Thought (CoT) prompting is used to facilitate multi-step reasoning. To balance coverage and computational cost, each dataset is subsampled to 250 items, and 10 generations are sampled per prompt, resulting in a total of $685,000$ long form responses across models and datasets evaluated against each of the 4 selected methods (see Section A.4).

## 7.3 KEY FINDINGS

**Verbalized Uncertainty** Our results show that models overwhelmingly defaulted to a small set of uncertainty scores (see Figure 4a), with higher confidences dominating the responses. This leads to a biased distribution of confidence scores, potentially driven by training data and instruction tuning. Calibration plots reveal no meaningful correlation between these verbalized scores and answer accuracy in all instruction tuned models, indicating that Verbalized Uncertainty is not a reliable proxy for true model confidence. This is slightly improved in reasoning models, most notably in the *gpt-oss* model family, providing scores that better correlate with answer accuracy on challenging datasets such as GPQA and SciBench (see Figure A.10). As no architectural or scale-related factors besides

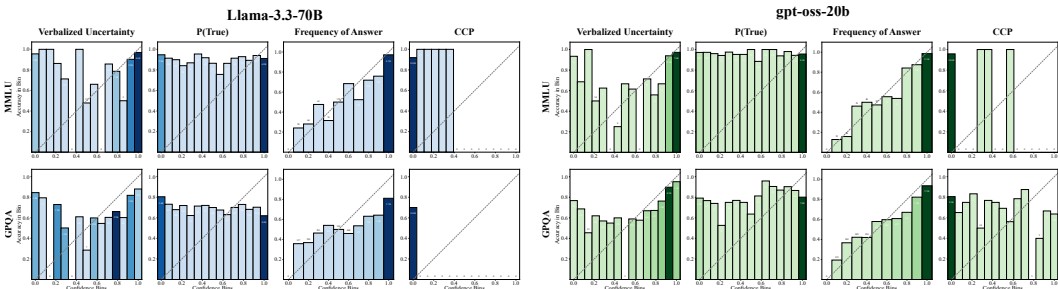

Figure 3: **Selected Calibration Plots For The Four Selected Methods.** Results for *Llama-3.3-70B* (left) and *gpt-oss-20b* (right) are shown, each for MMLU and GPQA for the four computed sequence level uncertainty methods. The full plots for all models and datasets per UQ method can be found in Section A.8.3.

the reasoning process clearly distinguish these models from others, the reason for the reliability is likely driven by training pipelines, which we cannot investigate due to them being undisclosed.

**P(True)** We find that P(True) suffers from pronounced response bias. In several models P(True) overwhelmingly assigns near 1.0 confidence with little use of intermediate confidence scores, resulting in a polarized distribution of certainty scores (see Figure 4b). This polarization may stems from the model's commitment to a single reasoning path before classification, and varies by model. Calibration plots further reveal no meaningful correlation between P(True) scores and actual correctness, indicating that P(True) cannot reliably quantify uncertainty in this setting.

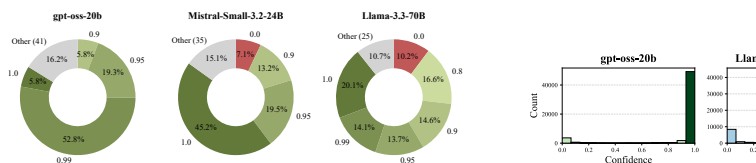

(a) **Verbalized Uncertainty** Confidence scores seen in less than $5\%$ of the total responses have been grouped into "Other", with the number of distinct confidence scores shown in brackets.

(b) **P(True)** Confidence scores are derived by probability mass assigned to labels (A) and (B), representing the model's confidence that the answer is true or false respectively.

Figure 4: **Distribution of Confidence Scores Across Selected Representative Models.** Confidence scores have been aggregated across all datasets ($57,500$ prompts in total). See Section A.8.4 and Section A.8.5 for extensive plots for all models.

**Frequency of Answer** Our evaluation (Figure 3) shows that higher answer frequencies strongly correlate with correctness across both multiple-choice and arithmetic tasks. More challenging datasets (e.g., GPQA, SciBench) exhibiting greater answer diversity, which reflects higher model uncertainty. Calibration plots confirm well-aligned confidence estimates based on Frequency of Answer, demonstrating the reliability of this approach. However, due to its computational cost, scaling with the number of samples per prompt, and its dependence on semantic clustering of outputs, it remains challenging or unapplicable to open-ended QA.

**Claim Conditioned Probability** In practice, we find that CCP suffers from vanishing sequence-level scores as generation length grows, i.e. multiplying a large number of token confidences drives overall confidence near 0. As a result, calibration plots (Figure 3) show no meaningful alignment with correctness. Furthermore, high impact of single NLI misclassifications and the inclusion of stop words in the aggregation further destabilize scores. These issues make CCP unreliable for sequence-level uncertainty estimation. But, its token-level insights could still inform targeted analyses once aggregation and domain-specific entailment are improved upon in future work.

## 8 Limitations

This study focuses on scientific QA in structured formats, which constrains the generalizability of the findings to other domains and task types such as open-ended generation or summarization. All datasets are in English, and the potential impact of cross-linguistic variation, input formatting, and prompt phrasing on calibration was not examined. Despite mitigation efforts using prompting strategies like APriCoT, the multiple-choice setting itself may introduce systematic biases, such as steering effects. Furthermore, the benchmark employs controlled inference conditions (e.g. fixed temperature, fixed decoding parameters) that may not capture the variability of real-world deployments. Finally, the evaluation of sequence-level uncertainty was limited to a subset of UQ methods with normalized outputs, selected to enable calibration analysis via calibration plots and summary metrics. This leaves unnormalized or claim-level metrics for future investigation into their potential value for detecting incorrect answers through high uncertainty estimates.

## 9 Conclusion

We presented a systematic evaluation of four UQ methods for LLMs across seven natural science datasets and up to 20 open-weight models, including base, instruction-tuned, and reasoning variants. We acquired our results by developing an open-source framework for benchmarking LLMs with a focus on efficiency and reproducibility. Expanding on existing literature and software frameworks, our contribution targets calibration instead of selective prediction to assess UQ method performance.

Our results reveal a pronounced polarization effect in token-level confidence distributions induced by instruction-tuning, which diminishes their utility as reliable uncertainty signals. Reasoning models are exposed to the same effect, but the reasoning process can mitigate this depending on its structure, as differences between providers are showing.

At the sequence level, we present results based on the evaluation of $685,000$ long form responses across models and datasets on 4 selected UQ methods representative for prominent approaches. We report that verbalized UQ methods exhibit consistently poor performance across most models. In particular, **P(True)** is adversely affected by the same confidence polarization previously observed at the token level. **Verbalized Uncertainty** scores are biased toward a narrow range of high-confidence scores and generally fail to correlate with answer correctness. Reasoning can improve on this, as seen in the *gpt-oss* family, underscoring the need for continued benchmarking to disentangle the impact of architectural design choices, fine-tuning data, and training paradigms on UQ behavior. The **Frequency of Answer** method based in semantic consistency showed strong reliability, albeit at high computational cost and with the non-trivial challenge of robust semantic equivalence detection. **Claim-Conditioned Probability (CCP)** suffers from vanishing sequence-level scores as generation length increases, compounded by NLI misclassifications and instability from stop-word aggregation.

We advocate that our results have substantial relevance outside of scientific QA tasks given the variety and number of datasets and models in use. As such, our findings highlight a critical need for the development of more robust, efficient, and theoretically grounded UQ methods for LLMs. Our results demonstrate that algorithmic advancements must always go alongside sustained empirical benchmarking to isolate the contributions of model architecture, fine-tuning strategy, and training data to uncertainty behavior in real world scenarios.

## 10 Future Work

Future research on uncertainty estimation in LLMs should explore several directions. Short-term efforts include extending benchmarks with new UQ methods, systematically studying inference parameters (e.g., temperature, sampling) and prompting strategies, and tracking UQ performance across emerging models to relate gains to architecture or training paradigms. Advancing UQ evaluation will also require new datasets that explicitly induce uncertainty or reformulate multiple-choice tasks into open-ended formats to isolate task-specific sources of uncertainty. Longer-term directions include claim-level uncertainty estimation, assessing reliability of individual statements or reasoning steps, and linguistic calibration, studying alignment between epistemic markers and model confidence.

REPRODUCIBILITY STATEMENT

Reproducibility in UQ for LLMs poses unique challenges due to the multistage evaluation pipelines and inherent randomness at each step, which can compound and introduce measurement noise. This stochasticity affects comparisons across UQ methods and model outputs, particularly when methods are evaluated on different generations, potentially contaminating results. To address these issues, we developed a modular benchmarking framework designed to ensure reproducibility and resource efficiency in large-scale LLM experiments. The framework supports extensible and replaceable computation nodes, caches probabilistic outputs such as model generations to ensure consistent re-evaluation, preserves intermediate outputs for qualitative inspection and allows incremental updates so that only affected steps need recomputation. It also enables sharing of intermediate results to reduce compute costs and allow other researchers to rerun the benchmark with their own methods. This is particularly valuable for ongoing research in UQ on models gated by proprietary access or costly hardware, supporting broader collaboration within the scientific community. These features not only enhance the reliability of the present study but also establish a foundation for future research in the field.

To facilitate reproducibility, the repository[2] includes the full framework `async-graph-bench` in its current form, final leaf-node outputs containing confidence scores required to reproduce the plots, a container definition file to build the execution environment, exact pip and driver versions, and instructions to install the framework and run the benchmarks presented in this work. While the framework is in its early stages, it is fully functional for reproducing the experiments reported here, and we plan to continue improving documentation and testing before its broader release in the near future.

LLM DISCLOSURE

LLMs were used in a limited manner to assist with writing, coding, and literature search. Specifically, LLMs provided minor support for language polishing, code-related assistance and identifying relevant references. All outputs were carefully reviewed, tested, and verified by the authors. LLMs did not contribute to the conceptualization of the research, the design of experiments, or the interpretation of results.

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

# A Appendix

## A.1 Repository

The repository accompanying this paper is available online at https://anonymous.4open.science/r/llm-uncertainty-bench-9B2B/.

## A.2 Model Selection

Table A.1: Overview of LLMs used in the experiments of this paper by provider, type, size, and release date. Model families of corresponding base, instruct and reasoning variants have been grouped together.

| Provider | Model Name | Type | Size | Release Date |
|---|---|---|---|---|
| OpenAI | gpt-oss-20b | Reasoning | 20B | Aug 2025 |
| | gpt-oss-120b | Reasoning | 120B | Aug 2025 |
| Mistral AI | Mistral-Nemo-Base-2407 | Base | 8B | Jul 2024 |
| | Mistral-Nemo-Instruct-2407 | Instruct | 8B | Jul 2024 |
| | Ministral-8B-Instruct-2410 | Instruct | 8B | Oct 2024 |
| | Mistral-Small-3.1-24B-Base-2503 | Base | 24B | Mar 2025 |
| | Mistral-Small-3.2-24B-Instruct-2506 | Instruct | 24B | Jun 2025 |
| | Magistral-Small-2507 | Reasoning | 24B | Jul 2025 |
| Meta LLaMA | Llama-3.1-70B | Base | 70B | Jul 2024 |
| | Llama-3.3-70B-Instruct | Instruct | 70B | Dec 2024 |
| | Llama-4-Scout-17B-16E | Base | 109B | Apr 2025 |
| | Llama-4-Scout-17B-16E-Instruct | Instruct | 109B | Apr 2025 |
| Qwen | Qwen3-30B-A3B-Base | Base | 30B | Jul 2025 |
| | Qwen3-30B-A3B-Instruct-2507 | Instruct | 30B | Jul 2025 |
| | Qwen3-30B-A3B-Thinking-2507 | Reasoning | 30B | Jul 2025 |
| DeepSeek AI | DeepSeek-R1-Distill-Llama-70B | Reasoning | 70B | Jan 2025 |
| | DeepSeek-R1-Distill-Qwen-32B | Reasoning | 32B | Jun 2024 |
| Google | gemma-3-27b-pt | Base | 27B | Mar 2025 |
| | gemma-3-27b-it | Instruct | 27B | Mar 2025 |

Please note that the model *Magistral-Small-2507* requires using a specific system prompt to enable reasoning[3]. We have tested the model without a system prompt (acting as an instruction tuned model) and with the system prompt provided, acting as a reasoning model. To highlight whether the reasoning behaviour was enabled through system prompt usage, we added a "Reasoning-Enabled" postfix to the model name in tables and plots.

### A.2.1 Model Configuration and Exceptions

All models were evaluated in their default configurations. No system prompts were employed, with the sole exception of *Magistral-Small-2507*, which requires a system prompt to enable reasoning prior to generating a final answer[4]. Without the system prompt, the model will behave like an instruction-tuned model. For the label-probability experiments, this model was tested in two variants: with reasoning enabled (*Magistral-Small-2507-Reasoning-Enabled*) and without reasoning (*Magistral-Small-2507*).

The base (pre-trained) model *gemma-3-27b-pt*, corresponding to the instruction-tuned *gemma-3-27b-it*, was excluded from the label-probability calibration experiments. In preliminary evaluations,

---

[3]https://huggingface.co/mistralai/Magistral-Small-2507

[4]see https://huggingface.co/mistralai/Magistral-Small-2506

the base model exhibited insufficient task comprehension, resulting in negligible probability mass assigned to label tokens within the top-20 most probable tokens. Consequently, label probabilities could not be retrieved through `vllm`, preventing the computation of confidence scores.

All models were used in their base configuration. With the exception of *Magistral-Small-2507*, no system prompts were used. *Magistral-Small-2507* uses the system prompt to elicit reasoning steps before providing a final answer. It was included in the label probability experiment both with (*Magistral-Small-2507-Reasoning-Enabled*) and without (*Magistral-Small-2507*) the system prompt. *gemma-3-27b-pt*, the pre-trained or base variant of *gemma-3-27b-it* was excluded from the experiment researching the label probability calibration due to lack in task comprehension. This resulted in no probability mass being assigned to label tokens within the 20 most probable tokens, rendering probabilities unaccessible via `vllm`. Therefore no certainties could be retrieved.

For the exact configuration parameters supplied to `vllm`, please refer to the models.py files located in the experiment subdirectories of the repository accompanying this paper.

## A.3 DATASET SELECTION

Table A.2: **Scientific QA datasets surveyed.** Selection criteria emphasized natural-science domains (especially physics) and inclusion of different reasoning requirements for answering, while providing a verifiable ground-truth.

| Dataset | Size | Task Format | Domain |
|---------|------|-------------|--------|
| **MMLU** | 15,908 | Multiple Choice | General (57 topics, including Physics) |
| **ARC** | 7,787 | Multiple Choice | Science (Physics, Chemistry, Biology, Earth Science) |
| **SciQ** | 13,679 | Multiple Choice | Science (Physics, Chemistry, Biology) |
| **GPQA** | 448 | Multiple Choice | Science (Physics, Chemistry, Biology) – Graduate-level |
| **GSM8K** | 8,792 | Arithmetic | Mathematics |
| **GSM-MC** | 8,787 | Multiple Choice | Mathematics |
| **SVAMP** | 1,000 | Arithmetic | Mathematics |
| **SciBench** | 2,229 | Arithmetic | Science (Physics, Chemistry, Biology, Medicine, Earth Science) |

For the exact dataset configuration parameters, please refer to the `data_source` subdirectories located in the experiment directories of the repository accompanying this paper.

## A.4 PROMPT COMPOSITION

The second experiment evaluates a total of $685,000$ long-form responses across models and datasets using sequence-level UQ methods. We include five multiple-choice QA datasets (MMLU, ARC-Easy, ARC-Reasoning, SciQ, GPQA) and three arithmetic QA datasets (GSM8K, SVAMP, SciBench).

For the MCQA datasets, we apply counterfactual prompting using the APriCoT strategy, evaluating each of the four answer options independently. This expands each item into four separate prompts. All datasets are subsampled to 250 items, and we sample 10 responses per prompt. Consequently, each arithmetic dataset yields 2,500 responses per model, and each MCQA dataset yields 10,000 responses per model, totaling 57,500 responses per model.

Due to high inference costs, we reduced the number of generated responses specifically for the *deepseek-ai/DeepSeek-R1-Distill-Llama-70B* model on the GPQA dataset by half, resulting in 5,000 evaluated responses for this model–dataset pair.

Across all 12 evaluated models, this results in a total of $685,000$ responses, each assessed individually using the four selected UQ methods in Experiment 2.

## A.5 Uncertainty Quantification Method Exclusion

Our selection of UQ methods reflects principled constraints: feasibility, interpretability, and applicability to realistic scientific reasoning tasks. An overview over exclusion of methods and short form explanation is provided in Table A.3. Only methods producing normalized sequence-level uncertainty scores are included in our analysis to enable reliability UQ validation by calibration. Subsequently, we focus our work on the following UQ methods: *Verbalized Uncertainty* (Tian et al., 2023), *P(True)* (Kadavath et al., 2022), *Frequency of Answer* (Wang et al., 2023), *Claim-Conditioned Probability (CCP)* (Fadeeva et al., 2024). More details of these methods are discussed in Section 7.1.

### A.5.1 Survey of UQ methods

We used the taxonomy from `LM-Polygraph` Fadeeva et al. (2023a) as a starting point and adapted the categorization of UQ methods into information-based, ensemble-based, density-based, reflexive or verbalized and meaning-diversity approaches. However, it is important to emphasize that `LM-Polygraph` targets selective generation, whereas our work focuses on calibration, the second core UQ task. Calibration requires normalized confidence scores in [0,1], which immediately excludes many of the 28 methods surveyed in `LM-Polygraph`. Thus, while we relied on Fadeeva et al. (2023a) for a broad overview of UQ approaches, the spectrum of methods applicable to calibration is necessarily narrower.

### A.5.2 Density-Based Methods

Density-based methods require access to the model's training data to estimate likelihoods or information-theoretic quantities. In our benchmark, the evaluated models are open-weight but not open-source, and their training corpora are unavailable. Consequently, density-based approaches cannot be applied, making them unsuitable for our calibration-focused evaluation.

### A.5.3 Ensemble-Based Methods

Ensemble methods combine multiple distinct models to estimate uncertainty. While potentially effective, they prevent the assessment of per-model calibration, which is central to our study. Additionally, ensembles introduce substantial design ambiguity regarding which combinations of models to use and incur significant computational costs. In contrast, the Frequency of Answer method we benchmark samples a single model multiple times, preserving per-model interpretability while allowing assessment of semantic consensus as an UQ signal.

### A.5.4 Claim-Level Methods

Semantic Entropy as a claim-level method was excluded due to methodological and computational limitations. Existing evaluations of this method are restricted to short-form factual QA datasets and do not assess calibration, offering limited evidence for their suitability in reasoning-intensive scientific QA. Furthermore, scientific answers frequently span hundreds or thousands of tokens. In our internal tests, claim-extraction prompts often produced tens to hundreds of trivial or irrelevant claims, each requiring separate LLM evaluation. This leads to computational costs far exceeding those of the Frequency of Answer method, our computationally most expensive method employed, making claim-level approaches impractical for large-scale scientific QA.

### A.5.5 Overview Over Excluded Methods

Table A.3: **Reasons for Exclusion of Uncertainty Metrics in Benchmark.** List of UQ Methods was adapted from the comprehensive evaluation and classification of uncertainty methods by Fadeeva et al. (2023a).

| Method | Category | Exclusion Reasons |
|---|---|---|
| Perplexity (Fomicheva et al., 2020) | Information-based | Produces unnormalized scores |

| Method | Category | Exclusion Reasons |
|---|---|---|
| Mean/max token entropy (Fomicheva et al., 2020) | Information-based | Produces unnormalized scores |
| Monte Carlo sequence entropy (Kuhn et al., 2023b) | Information-based | Produces unnormalized scores |
| Pointwise mutual information (PMI) (Takayama & Arase, 2019) | Information-based | Produces unnormalized scores |
| Conditional PMI (van der Poel et al., 2022) | Information-based | Produces unnormalized scores |
| Rényi divergence (Darrin et al., 2023) | Information-based | Produces unnormalized scores |
| Fisher-Rao distance (Darrin et al., 2023) | Information-based | Produces unnormalized scores |
| Focus (Zhang et al., 2023a) | Information-based | Produces unnormalized scores |
| Semantic entropy (Kuhn et al., 2023b) | Meaning diversity | Designed to work at the individual claim level rather than on entire sequences
Very high computational cost |
| TokenSAR (Duan et al., 2024) | Meaning diversity | Alters sentences in a way that violates autoregressive assumptions
Relies on NLI models whose performance in scientific contexts is inadequate |
| SentenceSAR (Duan et al., 2024) | Meaning diversity | Alters sentences in a way that violates autoregressive assumptions
Relies on NLI models whose performance in scientific contexts is inadequate |
| SAR (Duan et al., 2024) | Meaning diversity | Alters sentences in a way that violates autoregressive assumptions
Relies on NLI models whose performance in scientific contexts is inadequate |
| EigenScore (Chen et al., 2024) | Meaning diversity | Produces unnormalized scores |
| Sentence-level ensemble-based measures (Malinin & Gales, 2020) | Ensembling | Requires running multiple independent models, leading to high computational cost
Introduces extra variability that complicates comparison to single-model methods |
| Token-level ensemble-based measures (Malinin & Gales, 2020) | Ensembling | Requires running multiple independent models, leading to high computational cost
Introduces extra variability that complicates comparison to single-model methods |
| Mahalanobis distance (MD) (Lee et al., 2018) | Density-based | Density-based approach that requires proprietary training data |
| Robust density estimation (RDE) (Yoo et al., 2022) | Density-based | Density-based approach that requires proprietary training data |
| Relative Mahalanobis distance (RMD) (Ren et al., 2023) | Density-based | Density-based approach that requires proprietary training data |
| Hybrid Uncertainty Quantification (HUQ) (Vazhentsev et al., 2023) | Density-based | Density-based approach that requires proprietary training data |
| Number of semantic sets (NumSets) (Lin et al., 2023) | Meaning Diversity | Produces count-based outputs that are not normalized to [0,1] |
| Sum of eigenvalues of the graph Laplacian (EigV) (Lin et al., 2023) | Meaning Diversity | Produces unnormalized scores |
| Degree matrix (Deg) (Lin et al., 2023) | Meaning Diversity | Produces unnormalized scores |
| Eccentricity (Ecc) (Lin et al., 2023) | Meaning Diversity | Produces unnormalized scores |
| Lexical similarity (LexSim) (Fomicheva et al., 2020) | Meaning Diversity | Relies on surface-level token overlap instead of semantic meaning |
| Kernel Language Entropy (Nikitin et al., 2024) | Meaning Diversity | Produces unnormalized scores |
| LUQ ((Zhang et al., 2024a)) | Meaning diversity | Produces unnormalized scores |

## A.6 LABEL PROBABILITY CALIBRATION EXPERIMENT DESIGN

For the Label Probability Calibration Experiment, preliminary tests revealed that base models struggled with task comprehension: in many cases, none of the valid label tokens (A/B/C/D) appeared among the top-20 first-token predictions. To address this, we designed four prompt variants with different structural formats to identify the formulation that maximizes task comprehension. We define task comprehension as the average total probability mass assigned to the label tokens (A/B/C/D) in the model's first generated token.

The four prompt designs are presented in Section A.6.1. An overview of task comprehension scores for each prompt is provided in Section A.6.2, where Prompt 1 emerges as the most effective. Nevertheless, even with improved task comprehension, a substantial fraction of responses remained invalid (i.e. no extractable probability assigned to any label), as detailed in Section A.6.3.

We analyze the causes of these invalid responses in Section A.6.4 and outline the mitigation strategies applied to reduce them in Section A.6.5. Finally, we report the updated invalid-answer statistics for Prompt 1 after applying these adjustments in Section A.6.3. We report the distribution of token counts for the reasoning chains for reasoning models in Section A.6.7.

### A.6.1 PROMPT DESIGNS

Listing 1: **Prompt Design 1 for Experiment 1.** The prompt design uses 3-shot prompting. `<QUESTION>` and `<ANSWER CHOICE X>` are replaced with the individual questions and answer choices from the benchmarked dataset. Prompt Design 1 features markdown construct.

```
1  You are a highly capable language model trained for multiple-choice
   question answering.
2  Below are three examples of multiple-choice questions with labeled answer
    choices. Each example includes the correct answer.
3  After the examples, you will be given a new question with four labeled
   answer choices (A, B, C, D).
4
5  Your task is to select the answer choice you believe is correct by
   responding with only the corresponding label: A, B, C, or D.
6  Do not include any explanation or additional text.
7
8  ### Example 1:
9  **Question:** What is the capital of France?
10 A) Berlin
11 B) Madrid
12 C) Paris
13 D) Rome
14
15 **Correct Answer:** C
16
17 <Two More Examples Omitted for Readability>
18
19 ---
20
21 **Question:** <QUESTION>
22 A) <ANSWER CHOICE A>
23 B) <ANSWER CHOICE B>
24 C) <ANSWER CHOICE C>
25 D) <ANSWER CHOICE D>
26
27 **Correct Answer:**
```

Listing 2: **Prompt Design 2 for Experiment 1.** The prompt design uses 3-shot prompting. `<QUESTION>` and `<ANSWER CHOICE X>` are replaced with the individual questions and answer choices from the benchmarked dataset. Prompt Design 2 features no introductory text, role or task description. The format is designed to represent natural language without special formatting.

```
1  Question: What is the capital of France?
```

```
2  A) Berlin
3  B) Madrid
4  C) Paris
5  D) Rome
6
7  The correct answer is C
8
9  <Two More Examples Omitted for Readability>
10
11 Question: <QUESTION>
12 A) <ANSWER CHOICE A>
13 B) <ANSWER CHOICE B>
14 C) <ANSWER CHOICE C>
15 D) <ANSWER CHOICE D>
16
17 The correct answer is
```

Listing 3: **Prompt Design 3 for Experiment 1.** The prompt design uses 3-shot prompting. <QUESTION> and <ANSWER CHOICE X> are replaced with the individual questions and answer choices from the benchmarked dataset. The format of Prompt 3 is designed to represent natural language without special formatting. The format of the answer specifically requests the label, not the answer in general.

```
1  You are a highly capable language model trained for multiple-choice
   question answering. In the following examples, you will see questions
   with answer choices. The answer choices are preceded by the phrase "
   Answer Choices:". Each answer choice is annotated with one of the labels
   A, B, C or D. The correct answer to the question is given by the sentence
    "The label of the correct answer choice is" followed by the
   corresponding label. Your task is to answer the new question in the same
   format, outputting only the label of the correct answer to the question
   you are provided. Do not output anything other than one of the labels A,
   B, C or D.
2
3  Question: What is the capital of France?
4  Answer Choices:
5  A) Berlin
6  B) Madrid
7  C) Paris
8  D) Rome
9
10 The label of the correct answer choice is C
11
12 <Two More Examples Omitted for Readability>
13
14 Question: <QUESTION>
15 Answer Choices:
16 A) <ANSWER CHOICE A>
17 B) <ANSWER CHOICE B>
18 C) <ANSWER CHOICE C>
19 D) <ANSWER CHOICE D>
20
21 The label of the correct answer choice is
```

Listing 4: **Prompt Design 4 for Experiment 1.** The prompt design uses 3-shot prompting. <QUESTION> and <ANSWER CHOICE X> are replaced with the individual questions and answer choices from the benchmarked dataset. Prompt Design 2 features special tags to mark the answer given as a label.

```
1  You are a highly capable multiple-choice question answering model. Below
   are three examples that show the format you must follow. Each question
   has four answer choices labeled A, B, C, and D. Your task is to answer a
   new question by outputting the correct answer in the following format: <
```

```
ANSWER>X<ANSWER>, where X is the label corresponding to the correct
answer, A, B, C or D. Do not add any extra text or explanation.

Example 1:
Question: What is the capital of France?
Answer Choices:
A) Berlin
B) Madrid
C) Paris
D) Rome

<ANSWER>C<ANSWER>

<Two More Examples Omitted for Readability>

Now, please answer the following question in the same format.

Question: <QUESTION>
Answer Choices:
A) <ANSWER CHOICE A>
B) <ANSWER CHOICE B>
C) <ANSWER CHOICE C>
D) <ANSWER CHOICE D>

<ANSWER>
```

### A.6.2 TASK COMPREHENSION PER PROMPT DESIGN

Table A.4: **Task Comprehension Measured by Probability Mass Assigned to Answer Labels.** Mean over the sum of label probabilities per question for base, instruction-tuned, and reasoning models. On average, task comprehension is highest under Prompt 1.

| Model Category | Prompt 1 | Prompt 2 | Prompt 3 | Prompt 4 |
|---|---|---|---|---|
| Base Models | 0.2368 | 0.5928 | 0.4135 | 0.1632 |
| Instruction-Tuned Models | 0.9912 | 0.3597 | 0.9561 | 0.2646 |
| Reasoning Models | 0.7002 | 0.0895 | 0.4475 | 0.0022 |
| **Average** | **0.6427** | **0.3474** | **0.6057** | **0.1434** |

### A.6.3 INVALID ANSWER COUNT

Although we selected the prompt that yielded the highest task comprehension for our benchmark, the original experiment exhibited a substantial number of invalid answers, varying with prompt design. These invalid outputs contaminated the results. After performing analysis on the cause of this behaviour and employing mitigation strategies, we were able to significantly decrease the amount of invalid answers, as detailed in Table A.5.

Table A.5: **Number of Invalid Answers Given by Models for Different Prompt Designs Across All Datasets** ($n = 25{,}316$)**.** Invalid answers assign no probability mass to any answer-choice labels. This phenomenon is discussed above. Using the prompt design yielding the best task comprehension (as discussed in Section A.6.2), we adjusted the experiment to use structured decoding and increased the maximum token generation limit. After this adjustment, invalid answers may only occur when reasoning chains exceed 10240 tokens. The result is shown in column "Prompt 1 (SD)", drastically decreasing the amount of invalid answers.

| Model | Prompt 1 | Prompt 2 | Prompt 3 | Prompt 4 | Prompt 1 (SD) |
|---|---|---|---|---|---|
| gpt-oss-20b | 307 | 3356 | 498 | 25304 | 395 |
| gpt-oss-120b | 38 | 3323 | 27 | 24873 | 0 |
| Ministral-8B-Instruct-2410 | 0 | 0 | 0 | 0 | 0 |
| Mistral-Nemo-Base-2407 | 0 | 0 | 12574 | 0 | 0 |
| Mistral-Nemo-Instruct-2407 | 0 | 0 | 0 | 0 | 0 |
| Mistral-Small-3.1-24B-Base-2503 | 0 | 0 | 0 | 0 | 0 |
| Mistral-Small-3.2-24B-Instruct-2506 | 0 | 0 | 0 | 0 | 0 |
| Magistral-Small-2507 | 0 | 0 | 0 | 0 | 0 |
| Magistral-Small-2507-Reasoning-Enabled | 8604 | 18716 | 4749 | 21232 | 123 |
| Llama-3.1-70B | 0 | 13 | 0 | 6636 | 0 |
| Llama-3.3-70B-Instruct | 0 | 2527 | 19 | 125 | 0 |
| Llama-4-Scout-17B-16E | 2 | 3 | 0 | 5048 | 0 |
| Llama-4-Scout-17B-16E-Instruct | 0 | 15 | 0 | 0 | 0 |
| Qwen3-30B-A3B-Base | 2 | 1 | 0 | 55 | 0 |
| Qwen3-30B-A3B-Instruct-2507 | 0 | 6270 | 0 | 722 | 0 |
| Qwen3-30B-A3B-Thinking-2507 | 6242 | 11746 | 8656 | 21840 | 58 |
| DeepSeek-R1-Distill-Llama-70B | 425 | 7538 | 725 | 7477 | 148 |
| DeepSeek-R1-Distill-Qwen-32B | 914 | 7465 | 1170 | 3337 | 243 |
| gemma-3-27b-pt | 25316 | 25316 | 25316 | 25316 | 0 |
| gemma-3-27b-it | 0 | 431 | 0 | 2 | 0 |

### A.6.4 ANALYSIS OF INVALID ANSWERS

Our investigation identified two primary causes of invalid responses:

1. **Missing answer labels within top token alternatives:** Models sometimes did not generate any of the expected labels (A/B/C/D) within the top 20 token alternatives that vLLM provides per token. This occurred in:

   (a) *Base models*, which lacked sufficient task comprehension and failed to include any of the answer labels in the top 20 token alternatives.

   (b) *Reasoning models*, which faced the additional challenge of adhering to the expected output format after generating long-form reasoning, as the attention mechanism favours more recent tokens. Reasoning of several thousand tokens appeared to negatively effect task comprehension, leading to invalid answers in extreme cases.

2. **Overflowing generations:** Reasoning models produced outputs exceeding the originally set maximum token limit (4096 tokens), appearing to be stuck in prolonged reasoning cycles without converging to a final answer.

A special case is *Magistral-Small-2507*, which has reasoning enabled via the system prompt provided by MistralAI[5] . For this model, we consistently observed non-converging generations that produced a very high number of invalid answers, even on the relatively simple MMLU dataset compared to GPQA. Increasing the maximum token generation did not resolve the issue – the model still consistently hit the upper generation limit. Our solution for handling these invalid outputs is detailed in Section A.6.6.

---

[5]https://huggingface.co/mistralai/Magistral-Small-2506

### A.6.5 MITIGATION STRATEGIES

We addressed these issues as follows:

- Leveraged `vLLM`'s structured decoding feature to enforce compliance with the expected output format after reasoning using the regex `"[ABCD]"`.
- Increased the maximum token limit for reasoning from 4096 to 10240 tokens, a $2.5\times$ increase.
- Adjusted the generation of *Magistral* to handle infinite generations, as detailed in Section A.6.6.

While structured decoding enforces compliance with the expected output format, the prior evaluation of task comprehension per prompt remains essential. Forcing task compliance without adequate comprehension would result in outputs that, although syntactically valid, are semantically incorrect or inconsistent with the question, thereby undermining the reliability of uncertainty quantification and downstream analyses.

Based on the prior evaluation, we chose Prompt 1 for the updated experiment and discarded the other Prompt Designs.

### A.6.6 MAGISTRAL-SMALL-2507 INVALID ANSWER SOLUTION

To evaluate *Magistral-Small-2507* on label probability calibration and score distribution despite persistent infinite-generation issues, we first set the maximum token limit for reasoning chain generation to 4096 tokens. Generated outputs almost always contained unfinished reasoning chains, indicated by a missing end-of-reasoning token `"[/THINK]"`. We removed `"[/THINK]"` if present, trimmed any trailing incomplete sentence, and appended `".  I should now respond with the single label A, B, C, or D associated with the answer I consider most correct.[/THINK]"` to artificially terminate the chain and prompt a single-label response. The model was then queried again with this reasoning chain as a prefix, and label probabilities were extracted from the next generated token. Although structured decoding cannot be applied in this prefix-prompting setup, this procedure reduced invalid answers from 8604 to 123, as shown in Table A.5.

### A.6.7 REASONING TOKEN COUNT DISTRIBUTION FOR REASONING MODELS

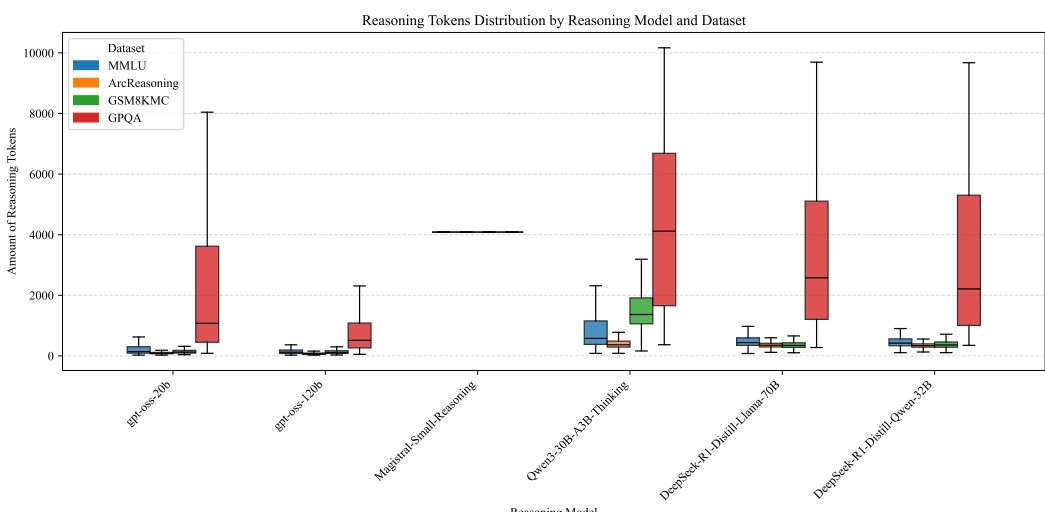

Figure A.1: **Distribution of Reasoning Tokens Produced by Models.** The figure displays a grouped boxplot illustrating the distribution of the Amount of Reasoning Tokens generated by different Reasoning Models across the individual Datasets. Models are grouped along the X-axis, with each boxplot representing the token count distribution for a specific dataset within that model. The central line in each box represents the median, and the box edges represent the first and third quartiles.

## A.7 LABEL PROBABILITY EXPERIMENT RESULTS

### A.7.1 COMPREHENSIVE PLOTS PER PROMPT WITH TABLES

Detailed calibration plots of label probabilities for all prompt designs and configurations are available in the project repository at `label_prob_calibration/resources/figures/full_plots` for the initial prompt designs without structured decoding and at `label_prob_calibration/resources_struct_decoding/figures/full_plots` for the chosen Prompt 1 and structured decoding enabled. Each file follows the naming convention `cal_plot_prompt<id>_table<t>_chosenonly<c>_norm<n>.svg`, where the placeholders encode the following settings:

- `prompt`: Identifier of the prompt design used to generate the plot (see Section A.6.1).
- `table`: Indicator of whether a table summarizing key calibration statistics is included.
- `chosenonly`: Flag specifying whether confidence scores are computed for all candidate labels (0) or restricted to the chosen (most probable) label (1).
- `norm`: Flag denoting whether label probabilities are normalized such that their sum equals one across all candidate labels.

In the following, the plots showing the calibration plots for Prompt 1 (best task comprehension) can be seen.

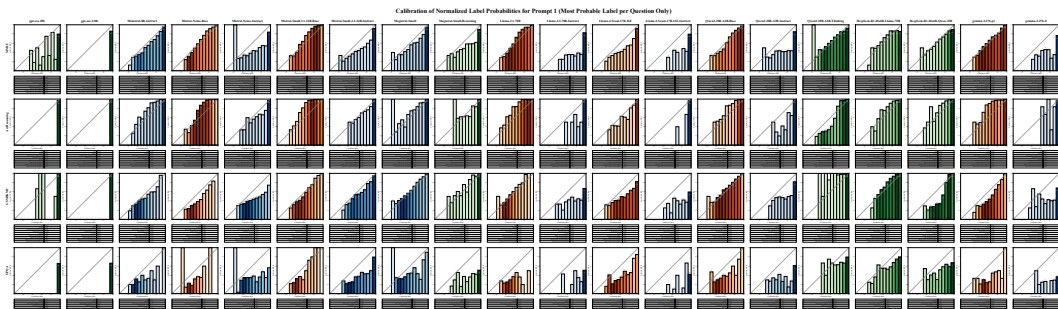

Figure A.2: **Calibration Plots for Prompt 1 with structured decoding enabled, using only the most probable label.** The columns represent different models, while the rows represent datasets. Tables below the plots list summary metrics such as ECE, normalized entropy of bucket counts and AUROC. Base models are shown in orange, instruction-tuned models in blue, and reasoning models in green. Darker colors indicate a higher number of items in the bin.

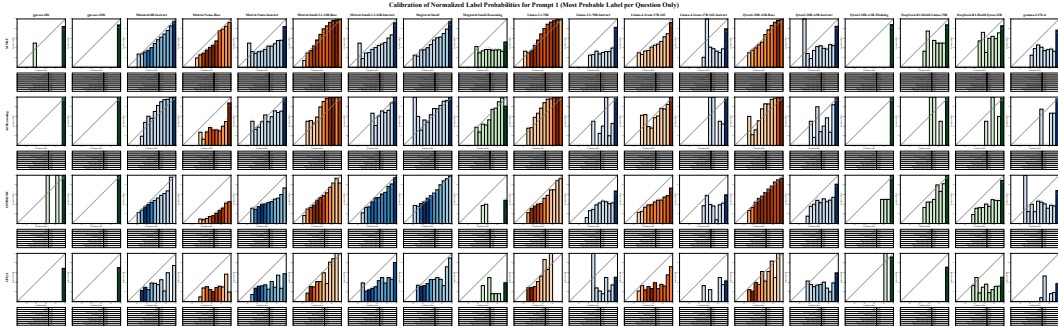

Figure A.3: **Calibration Plots for Prompt 1, with structured decoding disabled, using normalized label probabilities and only the most probable label.** The columns represent different models, while the rows represent datasets. Tables below the plots list summary metrics such as ECE, normalized entropy of bucket counts and AUROC. Base models are shown in orange, instruction-tuned models in blue, and reasoning models in green. Darker colors indicate a higher number of items in the bin.

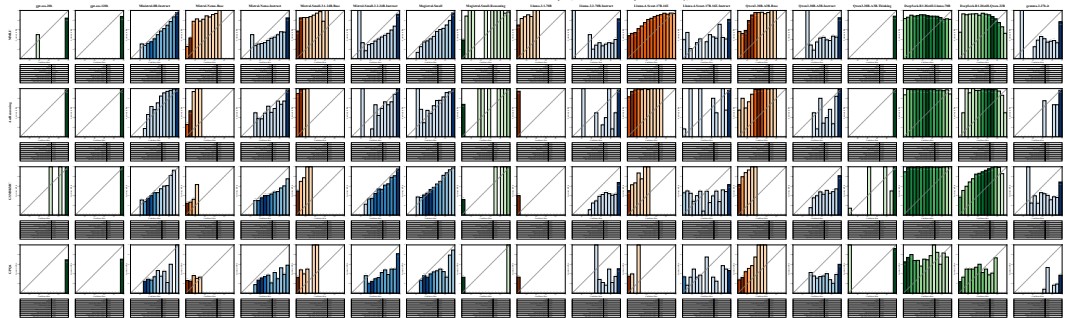

Figure A.4: **Calibration Plots for Prompt 1, with structured decoding disabled, using unnormalized label probabilities and only the most probable label.** The columns represent different models, while the rows represent datasets. Tables below the plots list summary metrics such as ECE, normalized entropy of bucket counts and AUROC. Base models are shown in orange, instruction-tuned models in blue, and reasoning models in green. Darker colors indicate a higher number of items in the bin.

### A.7.2 EFFECT OF NORMALIZATION

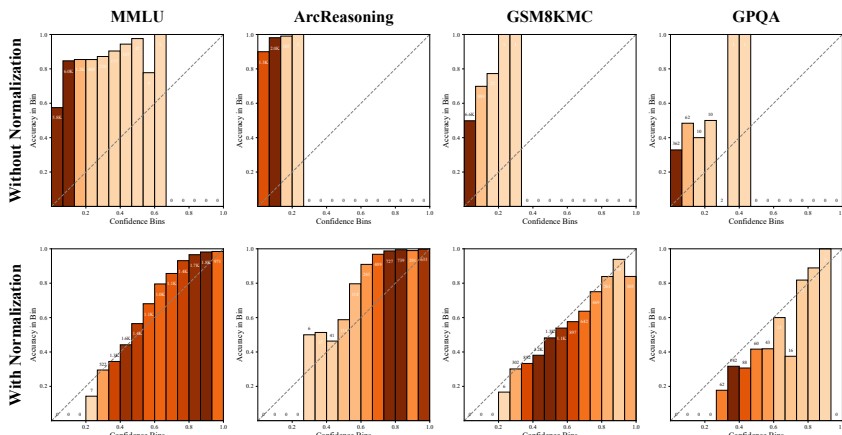

Figure A.5: **Representative Comparison of Calibration Plots for Unnormalized and Normalized Label Probabilities for the Model *Mistral-Small-3.1-24B-Base-2503* and Prompt 1 across all datasets used.** Calibration improves significantly after normalizing label probabilities.

While we had previously hypothesised that using raw label probabilities may enable the model to express general uncertainty, our results show that the task comprehension poses a bigger impact on the total probability mass assigned to label tokens. As a result, we find that raw probabilities are confounded by overall task comprehension and yield misleading calibration, while normalization (excluding non-label tokens) enables meaningful confidence estimates (see Figure A.5).

### A.7.3 SELECTION BIAS

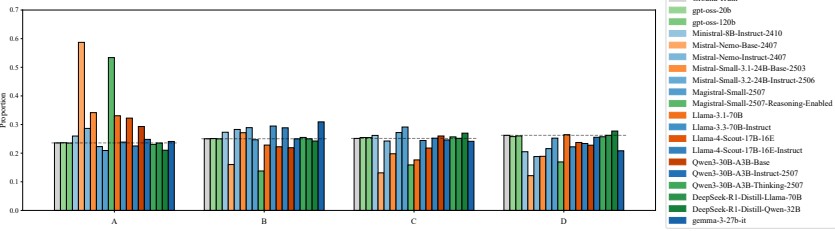

Figure A.6: **Probabilities for the Labels summed across all Datasets for each individual Model using Prompt 1.** The ground truth, represented by the distribution of the labels of the correct answers across all items in the datasets, is visualized by the grey bars and the dashed baselines.

### A.7.4 TASK COMPREHENSION PER PROMPT DESIGN

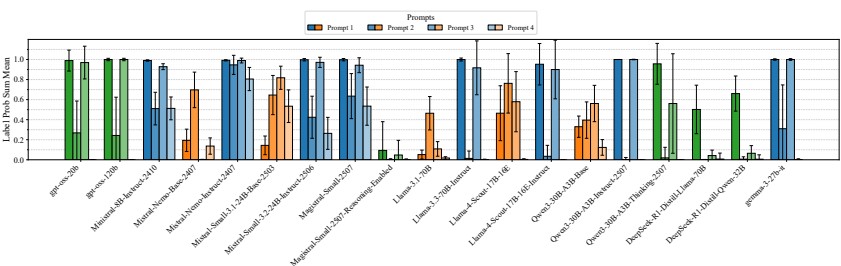

Figure A.7: **Task Comprehension per Prompt Design.** Mean over Sum of Label Probabilities for the Different Prompt Designs and Models. The data has been aggregated across all datasets, spanning 25,316 items per model and prompt design. Base models are shown in orange, instruction-tuned models in blue, and reasoning models in green.

### A.8 SEQUENCE-LEVEL CALIBRATION

### A.8.1 PROMPTS USED FOR QUESTION ANSWERING

The sequence-level experiments employed the APriCoT prompting strategy for MCQA and a standard Chain-of-Thought (CoT) approach for arithmetic question answering (Arithmetic QA), followed by final answer extraction. The exact prompts for both answer generation and final answer extraction are available in the repository accompanying this paper, specifically in `/llm-uncertainty-bench/seq_ue_calibration/nodes/apricot_mc_calc.py` for MCQA and `/llm-uncertainty-bench/seq_ue_calibration/nodes/arithmetic_calc.py` for Arithmetic QA.

### A.8.2 METRIC IMPLEMENTATION

**Verbalized Uncertainty** The prompt for Verbalized Uncertainty was adopted directly from the original work (Tian et al., 2023). The exact prompt can be found in `/llm-uncertainty-bench/seq_ue_calibration/run.py`, where it is provided to the corresponding computation node as `verbalized_prompt`.

**P(True)** The prompt used for the P(True) metric follows the formulation of (Kadavath et al., 2022). For APriCoT prompting in MCQA, a minor adaptation was applied, while preserving the core semantics of the original design. The exact prompts are available in `/llm-uncertainty-bench/seq_ue_calibration/nodes/ptrue.py`.

**Frequency of Answer** For this metric, 10 samples were generated per prompt. The *Frequency of Answer* of a given response is defined as the proportion of semantically equivalent answers within the set of 10 samples. Invalid answers are assigned a confidence of 0.0. Semantic equivalence was determined according to the task type:

- **Multiple-Choice QA:** Using the APriCoT prompting strategy, each option is independently evaluated, and the model classifies each option as correct or incorrect. Semantic equivalence is established when different generations reach the same classification decision for a given option.

- **Simple Arithmetic QA:** For datasets such as GSM8K, which primarily involve integers and rarely require floating-point precision, the final numeric result was extracted using a dedicated prompt and parsed into a numeric representation. Semantic equivalence is then determined by strict numeric equality of the extracted results. For details on the final answer extraction prompt, please refer to `/llm-uncertainty-bench/seq_ue_calibration/nodes/arithmetic_calc.py` in the repository.

- **SciBench:** This dataset presents additional complexity due to intricate computations and the inclusion of physical units, rendering the simple numeric matching used

for other arithmetic datasets insufficient. To address this, a specialized clustering prompt was developed to group sampled answers into semantically equivalent categories, with `Llama-3.3-70B-Instruct` serving as the judging model. Implementation details are provided in `/llm-uncertainty-bench/seq_ue_calibration/leaf_nodes/answered_correctly_scibench.py`.

For evaluation of the calibration of the Frequency of Answer metric, the binning strategy will be slightly modified. Unlike the other methods, this methods yields only discrete confidence values, determined by the number of sampled generations. With 10 generations per question, the resulting confidence scores can only take on values from 0.1 (indicating that all of the other nine sampled generations resulted in a different answer) to 1.0 (all generations produced the same result), in steps of 0.1. For responses that fail to yield a numeric outcome in arithmetic datasets, a confidence score of 0.0 is assigned. To accommodate these discrete confidence levels, 11 bins centered on the possible certainty scores will be used for generating the calibration plots and summary statistics thereof for the Frequency of Answer metric.

**Claim-Conditioned Probability (CCP)** The Claim-Conditioned Probability (CCP) metric, proposed by Fadeeva et al. (2024), was originally designed for claim-based uncertainty estimation. While conceptually valuable, applying CCP to long, complex generations proved challenging: extracting meaningful claims was computationally expensive, often unreliable, and complicated by interdependent claims that hindered aggregation. Nevertheless, CCP was included by aggregating token-level confidence scores through multiplicative composition. The implementation builds upon the authors' implementation of the metric in their `LM-Polygraph` framework (Fadeeva et al., 2023b) and was optimized for improved computational efficiency.

### A.8.3 Extensive Calibration Plots

In the following, extensive calibration plots for the evaluation of sequence level uncertainty methods are provided. Again, instruction tuned models are highlighted in blue, while reasoning models are highlighted in green. Darker shading indicates a higher number of items within each confidence bin.

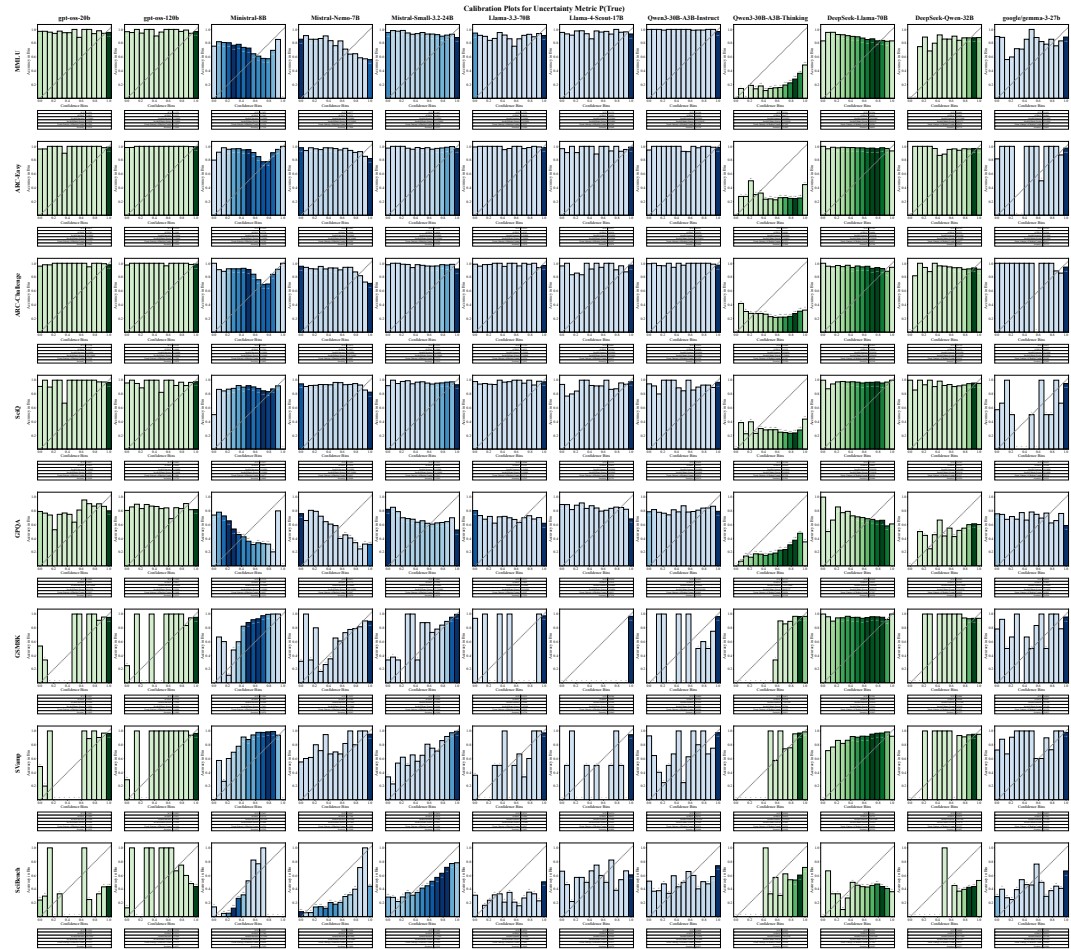

Figure A.8: **Calibration Plots for P(True).** Columns correspond to the models and rows to datasets.

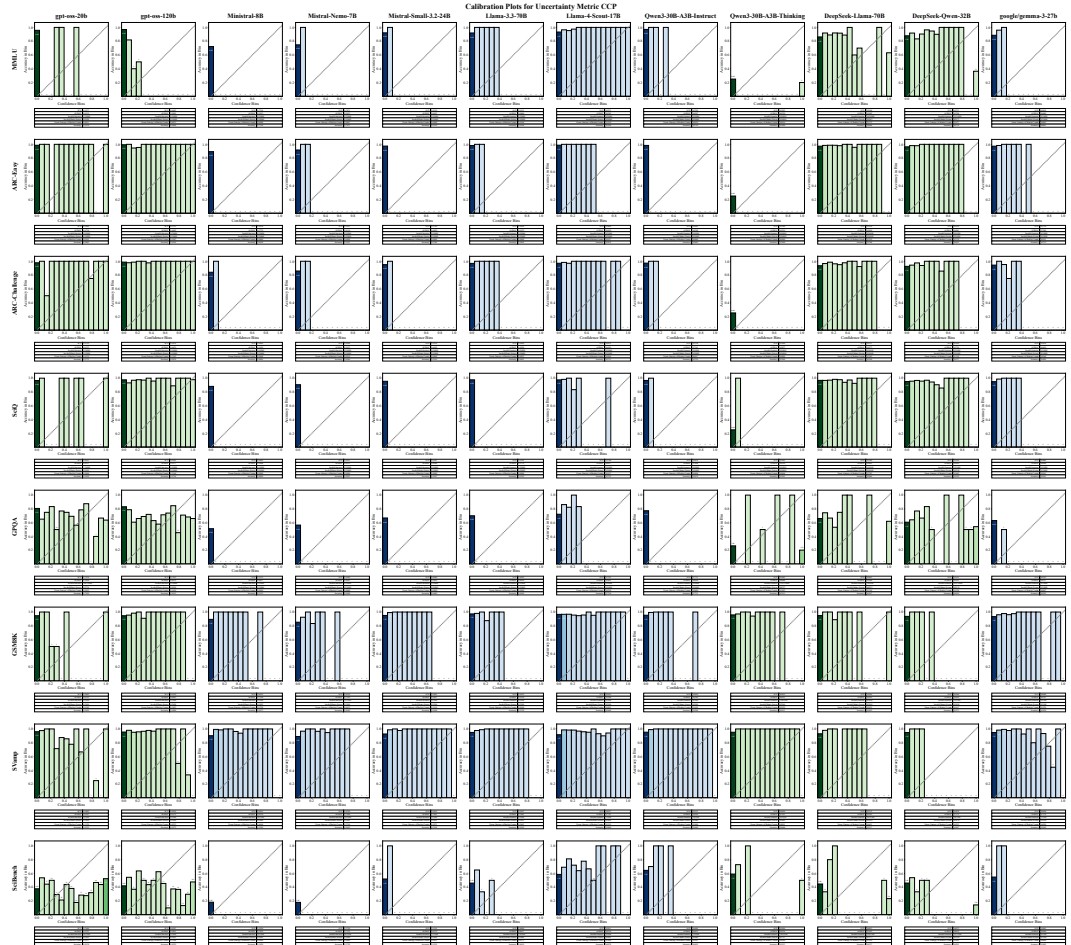

Figure A.9: **Calibration Plots for CCP.** Columns correspond to the models and rows to datasets.

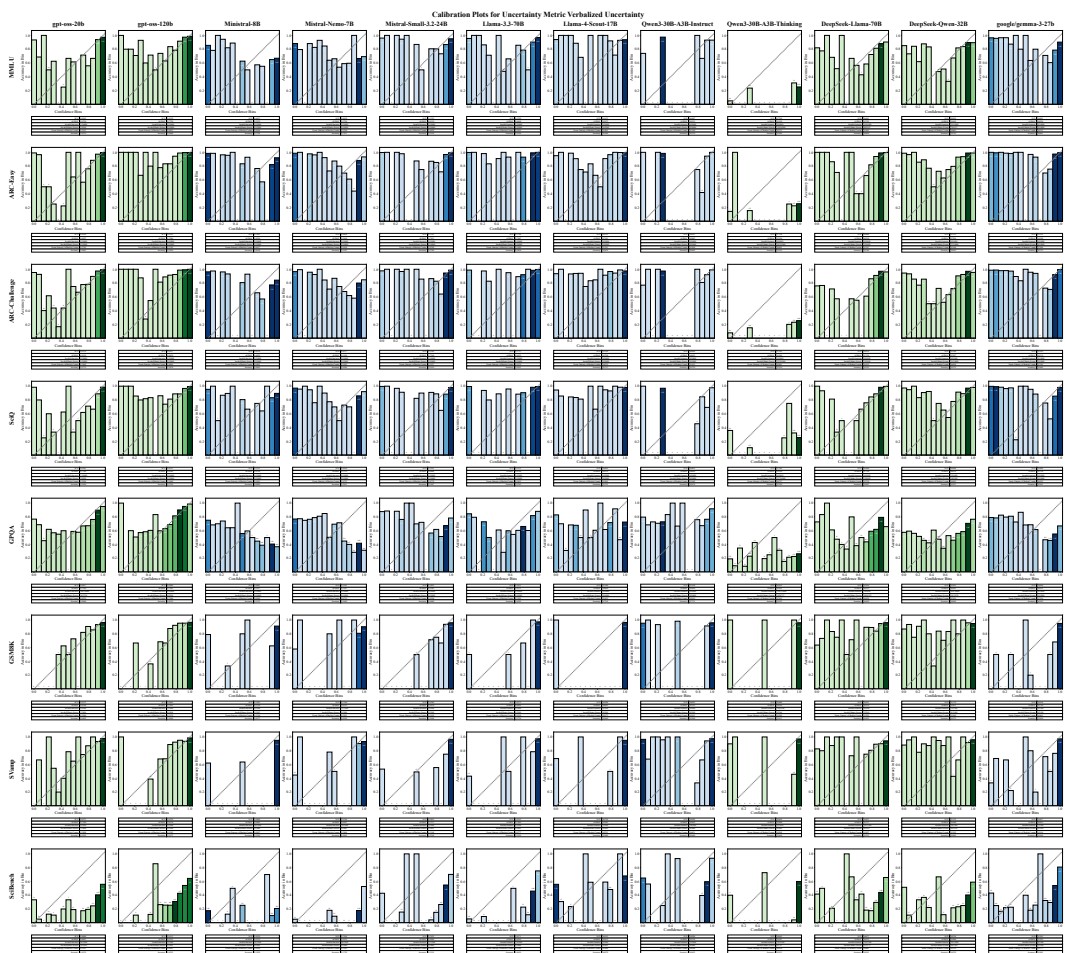

Figure A.10: **Calibration Plots for Verbalized Uncertainty.** Columns correspond to the models and rows to datasets.

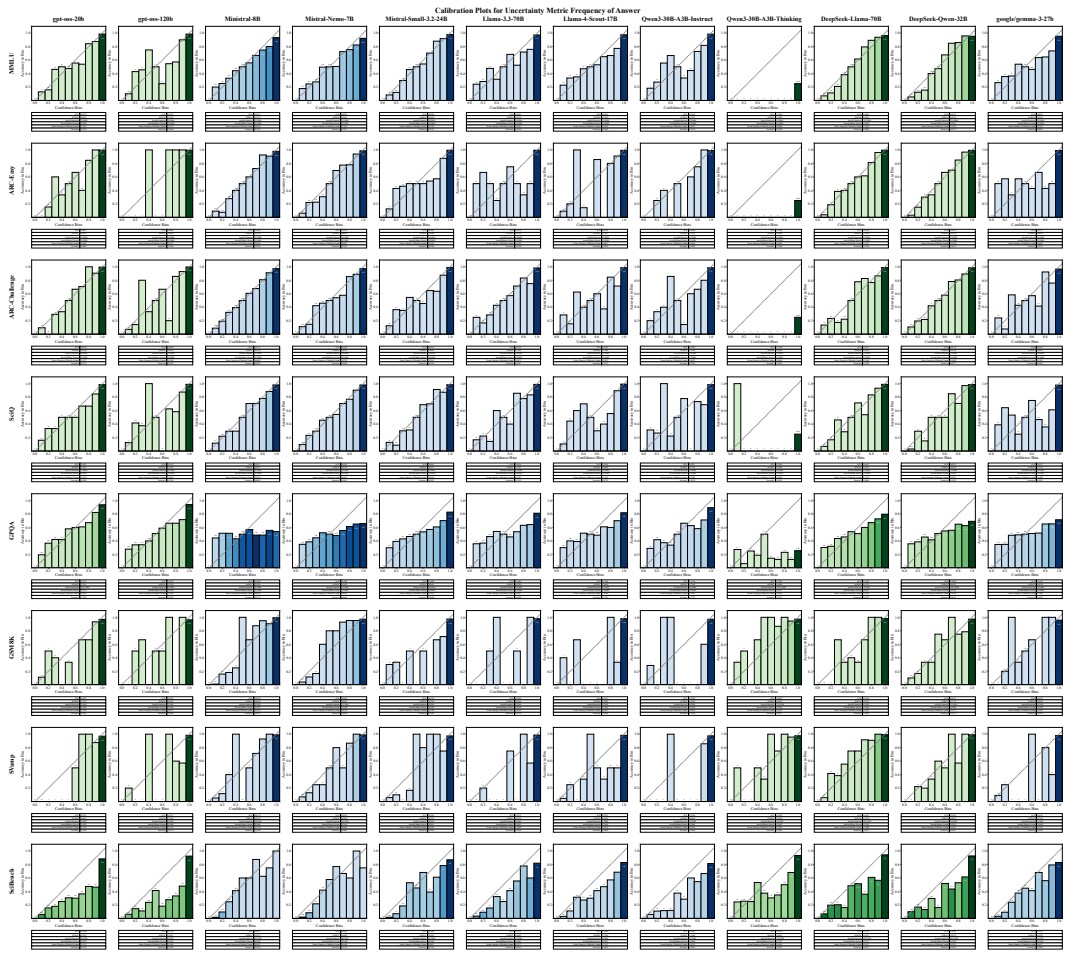

Figure A.11: **Calibration Plots for Frequency of Answer.** Columns correspond to the models and rows to datasets.

### A.8.4 VERBALIZED UNCERTAINTY CONFIDENCE SCORE DISTRIBUTION

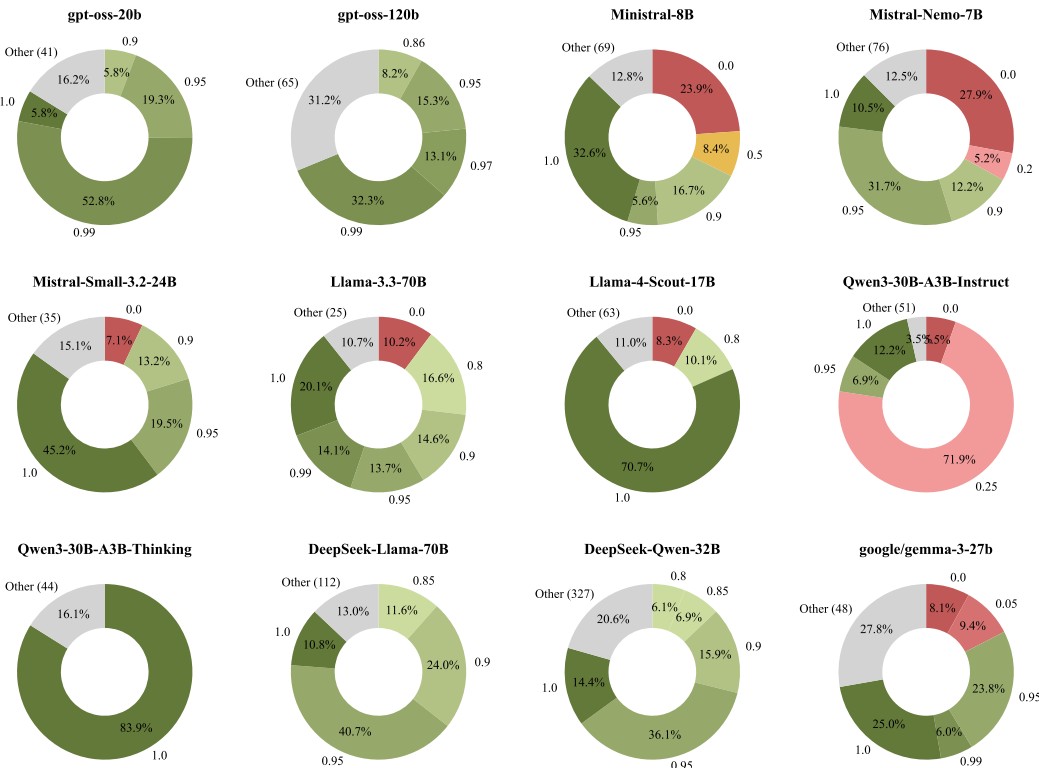

Figure A.12: **Distribution of Confidence Scores Provided During Verbalized Uncertainty Prompting Across Models.** Value counts are aggregated over all datasets (57, 500 prompts in total). Confidence scores seen in less than 5% of the total responses have been grouped into "Other", with the number of distinct confidence scores shown in brackets.

## A.8.5 P(TRUE) CONFIDENCE SCORE DISTRIBUTION

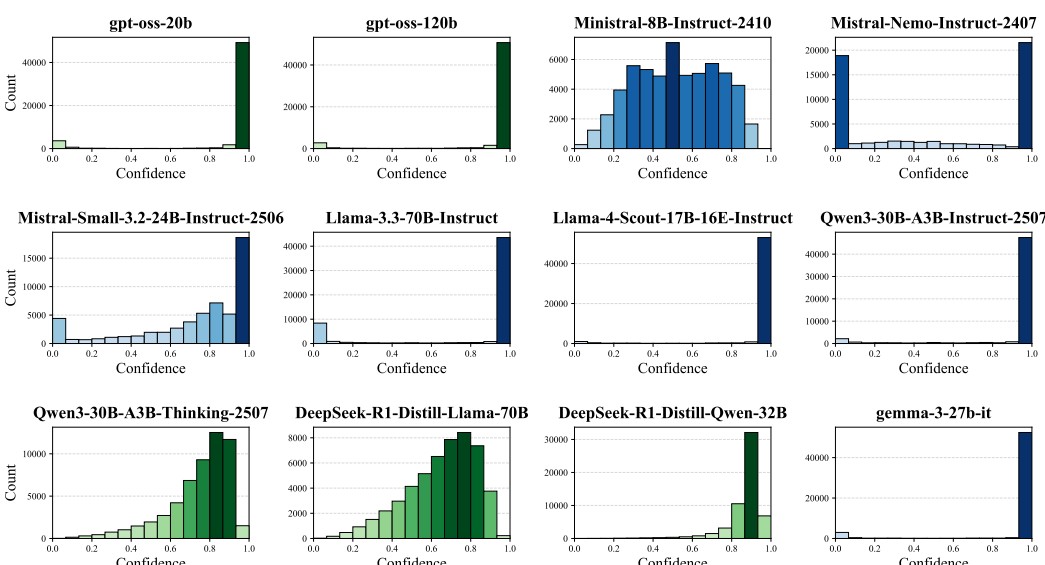

Figure A.13: **Distribution of Confidence Scores Assigned by P(True) Across Models.** Confidence scores have been aggregated across all datasets ($57,500$ prompts in total). Most models exhibit a clear polarization towards either (A) (representing the model's confidence that the answer is true) or (B) (representing the model's confidence that the answer is false) regarding the token probabilities.

## A.9 EFFECT OF TEMPERATURE

The following sections clarify why additional temperature-sweep experiments would not meaningfully affect our findings or improve the interpretability of the results.

### A.9.1 EXPERIMENT 1: TEMPERATURE IN TOKEN-LEVEL CALIBRATION

Experiment 1 evaluates calibration using the label probability of the first generated token under a fixed temperature of 1.0 (reasoning traces, when used, are decoded greedily at the same temperature). The central finding is a near-complete polarization of probability mass onto a single token, driven by extremely large logit margins. This collapse reflects a structural distributional issue, not a mild miscalibration. In such regimes, temperature scaling is ineffective: once margins grow this large, relative logit differences become dominated by noise and cease to carry calibration-relevant information. Adjusting temperature therefore cannot meaningfully "repair" the distribution or alter the calibration behaviour observed.

### A.9.2 EXPERIMENT 2: TEMPERATURE DURING RESPONSE GENERATION

In Experiment 2, temperature affects only the generative content of responses, not the utility of the evaluated UQ methods. Since our goal is not to optimize accuracy or reduce hallucinations, but to identify responses that exhibit uncertain or erroneous reasoning, additional generations at different temperature levels would not strengthen the benchmark. Running a full temperature sweep would require repeating $685,000$ generations per temperature value, with negligible methodological benefit: temperature primarily alters response variability, not the structure or comparability of the UQ scores themselves.

## A.10 EXPERIMENT 2: TEMPERATURE FOR VERBALIZED METHODS

For verbalized uncertainty methods, we use temperature 1.0 with greedy decoding. This ensures the model provides the reliability judgment it is most confident in. For P(True), the situation mirrors Experiment 1: the method depends on label probabilities, which frequently exhibit polarization

and show no reliable alignment with predictive accuracy. Increasing temperature cannot repair this mismatch, nor does it meaningfully alter calibration quality when the underlying distribution is already collapsed.

For these reasons, additional temperature-adjustment experiments would not yield meaningful benefits for either experiment.

