# OpenReview forum: "Benchmarking Uncertainty Estimation in Large Language Model Replies for Natural Science Question Answering"
_ICLR.cc/2026/Conference — Submitted to ICLR 2026_

### Official Review · Reviewer_nhTz · 2025-10-31

**Soundness:** 2
**Presentation:** 3
**Contribution:** 3
**Rating:** 4
**Confidence:** 3

**Summary:**

This paper presents a large-scale benchmark framework for evaluating uncertainty quantification (UQ) methods in scientific question answering (QA) with large language models (LLMs). The authors compare multiple token-level and sequence-level UQ methods (e.g., Verbalized Uncertainty, P(True), Frequency-of-Answer, and Claim-Conditioned Probability) across various open-source models and datasets. They also discuss the calibration behavior and reveal how instruction tuning or reasoning-style prompting may lead to probability polarization, thus weakening token-level confidence as a reliable UQ signal. Overall, the topic is important and timely, and the paper provides a broad and systematic experimental analysis.

**Strengths:**

1. Uncertainty quantification for LLM-based QA systems is highly relevant, particularly for scientific and high-stakes domains.
2. The benchmark spans multiple datasets and model families, including base, instruct, and reasoning variants, which allows a broad comparison of model calibration behaviors.
3. The identification of probability polarization after instruction-tuning and the empirical comparison of sequence-level methods (e.g., Frequency-of-Answer vs. CCP) provide useful insights for practitioners.
4. The paper emphasizes open-sourcing the benchmark and results, which can facilitate follow-up research and reproducibility within the community.

**Weaknesses:**

1.	Some core observations (e.g., why Mistral models behave differently or why GPT-based verbalized uncertainty performs better) are descriptive but not analytically explained. Additional ablation or controlled studies are needed to support these interpretations.
2.	The benchmark relies on prompts that maximize label-token probability, which introduces potential bias. The stability of results across different prompts, temperatures, or decoding settings is not sufficiently reported.
3.	The paper mentions that CCP scores often vanish due to sequence aggregation and NLI errors, but provides no mitigation or alternative aggregation strategies, limiting the method’s generalizability.
4.	Frequency-of-Answer is reported as the most reliable but computationally expensive method. The paper would benefit from a concrete compute/time/cost table to assess real-world feasibility.
5.	Many plots and tables lack statistical testing (e.g., bootstrapped CIs or significance analysis), making it unclear whether observed performance differences are robust.
6.	As a benchmark-oriented paper, it is important to clarify what the research community can concretely gain from this work beyond the empirical comparison. For instance, what types of new studies, analyses, or extensions could future researchers perform using the provided data or benchmark framework? Expanding this discussion would better articulate the lasting value of the benchmark to the community.

**Questions:**

1.	Could you include a cost analysis for the Frequency-of-Answer method (e.g., average sampling count, GPU-hours, runtime per instance)?
2.	Beyond performance comparison, what future research directions or applications does this benchmark enable for the community (e.g., evaluating new UQ techniques, cross-modal extensions, safety assessments)?
3.	How stable are the results under different prompts, temperatures, or decoding parameters? Would the key findings (e.g., performance ranking, polarization) remain consistent?
4.	Ensure that all abbreviations use capital initials and are defined upon first use, e.g., Large Language Models (LLMs), Question Answering (QA), etc.

---

> ### Author Response · Authors · 2025-11-20
> **W1 Mistral models’ behaviour / W2 Prompt stability and decoding options / W3 Vanishing CCP Scores**
>
> Thank you for your thoughtful review and for engaging with our work. We address the concern below and will reflect these clarifications in the revised manuscript.
>
> **W1 Mistral models’ behaviour**
>
> The differing behaviours we observe, including variation within the Mistral family and the stronger performance of GPT-based verbalized uncertainty (see figures A.6-A.9), most likely arise from differences in training pipelines, data composition, and model-internal tuning. As these details are proprietary, controlled ablations to isolate individual factors are not feasible.
>
> However, documenting these systematic differences remains valuable. Our benchmark provides the first broad, consistent comparison across multiple models of similar architectures, revealing stable patterns that merit deeper investigation by model developers. By releasing fully reproducible code and prompts, we enable researchers and model providers with access to internal training details to rerun and extend the benchmark, supporting more targeted analyses of the underlying causes.
>
> **W2 Prompt stability and decoding options**
>
> We agree that prompting introduces bias, which is unavoidable in LLM evaluation. In our benchmark, prompts optimized for label-token probability are used only for the label-probability experiments. To mitigate prompt bias, we test four distinct 3-shot prompt designs (see appendix section A.4.1) with minimal instruction content and report all results. Across models with different architectures and training pipelines, we observe consistent qualitative patterns under identical prompting, indicating stable and robust behaviour despite small shifts in absolute uncertainty scores.
>
> Regarding decoding settings, the label-probability experiments use temperature 1.0 without sampling (see section 8). The behaviour we report is not mild decoding sensitivity but a strong collapse of probability mass onto one label, with large logit margins. Post-hoc calibration methods, including temperature scaling, cannot meaningfully modify such distributions. Sampling-based strategies such as top-k or top-p are also irrelevant, since confidences are computed directly from token-level probabilities rather than generated samples.
>
> **W3 Vanishing CCP Scores**
>
> We agree that CCP’s instability under sequence aggregation and NLI errors raises important questions. However, the goal of our work is not to propose new UQ methods or improvements, but to provide a comprehensive and realistic benchmark of core approaches. Identifying where widely used methods fail is itself a central contribution, especially in a landscape where many new UQ techniques are published with (in our view) limited validation. This goal to highlight known or new problems aligns with the ICLR’26 reviewer guide, which emphasizes the importance of work that “Is to \[...\] address a known application or problem or draw\[s\] attention to a new application or problem.”
>
> Our findings highlight concrete failure modes that require further methodological development, and we see this as an opportunity for future work rather than a limitation of the benchmark. By making all code and evaluation pipelines reproducible, we enable researchers to explore alternative aggregation strategies on top of our framework.

---

> ### Author Response · Authors · 2025-11-20
> **W4 Frequency-of-Answer / W5 Statistical testing**
>
> **W4 Frequency-of-Answer**
>
> We appreciate the reviewer’s suggestion. Frequency of Answer (FoA) measures semantic consistency across multiple samples, and in our benchmark we use 10 samples per prompt. This implies a compute cost that is approximately ten times that of a single-query method, with SciBench requiring one additional lightweight LLM call for clustering arithmetic results that differ only by rounding. Thus, the computational cost scales directly with the number of samples, and future work could examine how calibration performance changes with fewer samples.
>
> FoA is feasible in our setting because all datasets have verifiable ground truth, allowing model outputs to be projected onto a single canonical answer for clustering. This projection is not possible for fully open-ended QA, which is why FoA is currently impractical for real-world deployment. However, evaluating FoA remains valuable: it provides insight into the reliability of semantic-consistency-based uncertainty estimation and can guide the development of more efficient successors that operate without requiring many samples or strict ground-truth formats.
>
> **W5 Statistical testing**
>
> We appreciate the suggestion. While statistical testing is valuable in many contexts, it is not standard in UQ evaluation for LLMs from our point of view. Our perspective on prior work typically shows that authors report point estimates on summary statistics like ECE, AUROC, or correlations without confidence intervals or hypothesis testing (e.g., see the GPT-4 Technical Report, LM-Polygraph paper, CCP article, Verbalized Uncertainty article). Our benchmark follows these (assumed) conventions to ensure comparability with existing literature. However, for this very reason we also acknowledge the argument here.
>
> Let us emphasize though that our conclusions do not depend on effects from individual datasets. The benchmark comprises 747,000 responses across 11 models from different providers, 8 datasets spanning fact retrieval, symbolic and scientific reasoning, arithmetic QA, and MCQA with APriCoT prompting, as well as multiple prompting variants and UQ methods. The strong consistency of patterns across these diverse settings provides robust empirical support for phenomena such as probability-mass polarization and systematic miscalibration, which are large in magnitude and highly reproducible.
>
> We agree that future work could incorporate bootstrapped confidence intervals or significance testing of fine-grained comparisons. Given the scale, diversity, and clear cross-setting stability of our results, we believe the current presentation is sufficient to support the reliability of our findings while remaining aligned with field norms.

---

> ### Author Response · Authors · 2025-11-20
> **Answers to the Questions Raised (Q1-Q3)**
>
> **Q1 Computational Effort**
>
> As noted in W4, the computational overhead of FoA scales linearly with the number of samples, while the verbalized approaches incur only minimal additional cost. CCP is more sensitive to caching efficiency due to repeated encoder comparisons. We agree that these distinctions should be made clearer and will revise the manuscript to more explicitly compare the computational requirements of all evaluated methods.
>
> **Q2 Future Directions/ Applications**
>
> Our benchmark evaluates core UQ approaches that form the basis for many subsequent and derivative methods. We show that discrepancies with prior work largely arise from (a) benchmarking in narrow or unrealistic domains and (b) limitations of widely used metrics such as ECE. Clarifying these issues provides important insights into the current shortcomings of UQ methods and offers guidance for evaluating new techniques. Because our framework is focussed on reproducibility and fully extensible, it can naturally support future methods as they emerge. Additional discussion of future research directions is provided in Section 10.
>
> **Q3 Prompt Design and Decoding Settings**
>
> We systematically examined prompt design, as detailed in Appendix A.4.1, and discussed decoding parameters explicitly in the limitations section (Section 10). If there are specific aspects you would like us to elaborate on beyond what is already included, we would greatly appreciate further clarification so that we can address them directly.
>
>
> We thank the reviewer for their constructive input and will consider the highlighted points in the revised manuscript. We hope that our responses and the forthcoming revisions will address the concerns raised and provide a clearer basis for positively evaluating the contribution of our work.

---

### Official Review · Reviewer_hmfZ · 2025-10-31

**Soundness:** 2
**Presentation:** 2
**Contribution:** 2
**Rating:** 2
**Confidence:** 5

**Summary:**

This paper presents an empirical benchmark evaluating various uncertainty quantification methods for LLMs' responses in natural science question answering. The authors conduct experiments across 11 LLM models and 8 scientific QA datasets. They identify several issues, including pronounced polarization of token probabilities and systematic biases in verbalized uncertainty, and find that answer frequency offers the most reliable calibration, despite its high computational cost.

**Strengths:**

- The paper's motivation to investigate uncertainty quantification specifically for scientific question answering is relevant.
- The authors attempt to evaluate a range of intrinsic and extrinsic uncertainty quantification methods, providing an empirical comparison across different approaches.
- This paper evaluates 11 models across 8 datasets spanning multiple task formats (MCQA and arithmetic QA), providing an empirical investigation that includes base, instruction-tuned, and reasoning model variants.

**Weaknesses:**

- While the authors claim this is the "first large-scale benchmark" for UQ in scientific QA, LM-Polygraph [1] already provides a more comprehensive evaluation framework with 28 UQ methods. Many datasets used here (MMLU, ARC, GSM8K, SVAMP) are already standard in existing UQ literature, undermining the claimed distinctiveness of the scientific QA focus.
- Despite emphasizing "Natural Science Question Answering" in the title, the paper's main findings, "probability polarization and verbalized uncertainty bias," can apply equally to general QA settings, with no meaningful analysis of what makes scientific reasoning unique.
- I found some critical issues were ignored in the main text. For instance, Table A.5 reveals that numerous models produce massive invalid answers under specific prompts (e.g., Magistral-Small-Reasoning has 18,716/25,316 invalid responses for Prompt 2). However, the main paper fails to acknowledge or discuss this severe reliability issue that fundamentally undermines the evaluation.
- The exclusion logic for some specific UQ methods appears arbitrary. Semantic Entropy is rejected for being "claim-level" despite its proven effectiveness across settings. At the same time, all ensemble/density-based methods are dismissed for "computational cost," even though Frequency of Answer (requiring 10 samples per prompt) is included, revealing a lack of principled selection criteria.
- About the probability polarization phenomenon, while the paper documents that instruction tuning induces strong probability polarization (evident in Figure 1), it remains a purely descriptive observation without investigating the underlying causes (training objectives? data distributions?), potential mitigation strategies (temperature scaling? calibration methods?), or whether the effect varies systematically across question types.
- I think the most intriguing finding of the paper is "extreme polarization in reasoning models (Magistral, Qwen-Thinking, DeepSeek-R1), with Qwen3-30B-A3B-Thinking placing nearly all predictions in the highest confidence bin". However, it receives only surface-level discussion, with no analysis of how reasoning chains affect uncertainty or whether better uncertainty estimates could be extracted from the reasoning process itself, as discussed in a set of recent works [2-6].
- Questionable Dataset Selection. The claim that physics serves as a "representative natural science domain" is unsubstantiated since "chemistry/biology" may exhibit different uncertainty patterns. Most datasets actually contain multi-disciplinary content rather than pure physics. GPQA's 448 samples are insufficient for a "large-scale" benchmark, and the exclusive focus on verifiable formats (MCQA/arithmetic) excludes open-ended scientific QA and genuine proof-based reasoning tasks, severely limiting generalizability.

[1] Fadeeva et.al, LM-Polygraph: Uncertainty Estimation for Language Models, EMNLP 2023.

[2] Mei et.al, Reasoning about Uncertainty: Do Reasoning Models Know When They Don't Know, arXiv 2025.

[3] Zhang et.al, CoT-UQ: Improving Response-wise Uncertainty Quantification in LLMs with Chain-of-Thought, ACL 2025.

[4] Becker et.al, Cycles of thought: Measuring llm confidence through stable explanations, arXiv 2024.

[5] Da et.al, Understanding the uncertainty of llm explanations: A perspective based on reasoning topology, COLM 2025.

[6] Mo et.al, Tree of uncertain thoughts reasoning for large language models, ICASSP 2024.

**Questions:**

Please see the weaknesses I've outlined.

---

> ### Author Response · Authors · 2025-11-20
> **W1 Comprehensiveness compared to LM-Polygraph**
>
> Thank you for your thoughtful review and for engaging with our work. We address the concern below and will reflect these clarifications in the revised manuscript.
>
> We thank the reviewer for raising this point and acknowledge the value of LM-Polygraph. Our work builds on its survey of UQ methods, but we maintain that our paper provides the first large-scale benchmark for calibration-focused UQ in scientific question answering. The key distinctions are as follows.
>
> **First, LM-Polygraph does not evaluate UQ in scientific QA.** Its QA experiments use only bAbI, a synthetic text-comprehension dataset with minimal reasoning complexity. In contrast, our benchmark spans eight datasets covering multiple-choice and open-ended scientific QA, including ARC Reasoning, SciBench and GPQA, which require multi-step, symbolic, and domain-specific reasoning. Standard datasets (MMLU, GSM8K, SVAMP) are included to ensure comparability with prior UQ work and to contextualize method behaviour across difficulty regimes.
>
> **Second, LM-Polygraph targets selective generation, whereas our work focuses on calibration**, the second core UQ task. Calibration requires normalized confidence scores, which exclude many of the 28 methods in LM-Polygraph. We will clarify this distinction explicitly in the method-exclusion section of the appendix in the revised version.
>
> **Third, our benchmark is far broader in scale and model diversity**. We evaluate a total 747,500 long form responses on 11 contemporary models (7B-70B) from five providers across eight datasets of varying difficulty levels. LM-Polygraph evaluates QA on a single simple dataset and two 7B models.
>
> **Finally, our study provides new insights not captured in prior work.** At token-level we investigate polarization of token probabilities previously only reported by one paper in a narrow setting. We show that verbalized approaches exhibit systematic bias and no correlation with correctness. In contrast, Frequency of Answer, a semantic-consistency approach, produces well-calibrated scores. We release all code, prompts, configurations, and raw uncertainty outputs, together with a novel benchmarking framework that ensures reproducibility and supports ongoing, open research in this field. We highlight that ECE is an unreliable measure of reliability of uncertainty scores on saturated QA benchmarks.
>
> **For these reasons, our benchmark offers the first systematic, calibration-focused evaluation of core UQ methods on realistic scientific QA tasks, addressing gaps not covered in LM-Polygraph or other existing studies.**

---

> ### Author Response · Authors · 2025-11-20
> **W2 Scientific QA Focus**
>
> We respectfully disagree that our findings are unrelated to scientific QA. As discussed in Section 3.1, scientific QA differs fundamentally from standard factual QA by requiring multi-step reasoning, symbolic manipulation, and domain-specific knowledge. These characteristics introduce uncertainty failure modes that are not observable in simple fact-retrieval datasets.
>
> Our benchmark reflects these challenges through datasets such as ARC Reasoning, SciBench and GPQA, which demand complex scientific and mathematical reasoning and stand in clear contrast to the low-complexity or saturated benchmarks commonly used in UQ research. On such saturated datasets, methods can appear reliable simply because models achieve high accuracy, as we highlight in this study.
>
> Additionally, the impact of hallucinations is substantially more consequential in scientific QA, where errors carry higher stakes and UQ is explicitly needed to assess reliability. Our results demonstrate that widely used UQ methods fail in these realistic scientific settings, a conclusion that cannot be drawn from existing benchmarks.

---

> ### Author Response · Authors · 2025-11-20
> **W3 Invalid Answers / W4 Selection Criteria for UQ Methods / W5 Probability Polarization Phenomenon**
>
> **W3 Invalid Answers**
>
> We thank the reviewer for raising this point and agree it warrants clearer discussion. Invalid responses arise in base models lacking instruction-following capabilities and reasoning models due to the challenge of adhering to the original prompt task specification after generating intermediate reasoning steps. The high invalid answer count in Magistral is simply a result of bad model performance on this challenge. The high number of invalid answers cited by the reviewer originates from a prompt variant we intentionally discarded. We evaluated four prompts and selected the one with the best task adherence across models. The reported invalid outputs therefore reflect a rejected prompt, not the one used in our main experiments. These failures do not undermine our evaluation but instead justify the need for careful prompt selection. While invalid responses cannot be completely avoided, we mitigated them as much as possible and will make this explicit in the revised manuscript.
>
> **W4 Selection Criteria for UQ Methods**
>
> We thank the reviewer for raising this point and will clarify the selection criteria in the revised manuscript.
>
> Our exclusion of density-based and ensemble-based methods is based on practical and methodological constraints. Density-based approaches require access to the model’s training data, which is unavailable for the open-weight (but not open-source) models we evaluate. Ensemble methods require combining multiple distinct models, which prevents assessing the calibration behaviour of individual models (central to our study) and introduces substantial design ambiguity in choosing ensemble configurations. Frequency of Answer samples a single model and therefore preserves per-model interpretability.
>
> Semantic Entropy was not excluded arbitrarily. It has only been evaluated on two short-form factual QA datasets and without any calibration analysis, offering limited evidence for its suitability in reasoning-intensive scientific QA. Its reliance on claim-level extraction also poses major scalability challenges: scientific answers can span hundreds or thousands of tokens, and in our internal tests the claim-extraction prompts produced many trivial or irrelevant claims, each requiring separate LLM evaluation. This results in computational costs far exceeding those of Frequency of Answer, making the method impractical for large-scale scientific QA.
>
> Our selection therefore reflects principled constraints of feasibility, interpretability, and applicability to realistic scientific reasoning tasks, and we will make this rationale explicit in the revision.
>
> **W5 Probability Polarization Phenomenon**
>
> Our benchmark aims to document empirical behaviour of UQ methods. We attribute the instruction-tuning induced probability polarization effect to training difference, however investigating this is not possible, as modern LLMs rely on proprietary training data, undisclosed fine-tuning pipelines, and closed objectives, preventing mechanism-level analysis for third-party researchers.
>
> Mitigation strategies such as temperature scaling are also ineffective: the phenomenon reflects an extreme collapse of probability mass with large logit margins, not a mild calibration error, and post-hoc calibration cannot meaningfully adjust such distributions.
>
> Our contribution is therefore to quantify the effect systematically across 11 models and 7 datasets and to show that it appears consistently across dataset types and question categories. This establishes its generality and provides the empirical basis for future work on mitigation or training-level solutions.

---

> ### Author Response · Authors · 2025-11-20
> **W6 UQ on Reasoning Paths / W7 Dataset Selection and Natural Science Coverage**
>
> **W6 UQ on Reasoning Paths**
>
> We thank the reviewer for highlighting this point. We do discuss the phenomenon and argue that reasoning models effectively commit to a single answer early in the reasoning chain. Their final certainty reflects confidence in the sampled reasoning trajectory rather than genuine uncertainty about the question. As a result, they produce highly polarized probabilities that do not account for alternative reasoning paths.
>
> Regarding the cited recent works, several are not peer-reviewed or are available only on arXiv. As stated in the ICLR 2026 Reviewer Guide, authors are not required to compare against non-peer-reviewed work.
>
> More importantly, many of the cited methods build on semantic consistency or verbalized uncertainty, both of which we evaluate directly in our benchmark. Our goal is not to introduce new methods but to provide a fundamental, large-scale analysis of core UQ approaches in realistic scientific QA, which forms the basis relevant to these derivative methods. Extending our study to design new UQ approaches is outside the scope of the benchmark.
>
> **W7 Dataset Selection and Natural Science Coverage**
>
> We thank the reviewer for these comments and acknowledge that our framing of science domains we benchmark was imprecise and we will revise the manuscript accordingly. In reality, 4 of our 7 datasets (SciBench, GPQA, SciQ, ARC Reasoning) contain questions from physics, chemistry and biology. The benchmark therefore spans interdisciplinary scientific reasoning, covering symbolic, numerical, factual, and multi-step tasks across multiple scientific domains. This supports the generality of our findings in the scientific domain.
>
> Regarding GPQA, its value lies in its expert-level, multi-step reasoning complexity rather than its size. Moreover, UQ is evaluated on model responses. Using the subsampling, prompting and 10 samples per prompt as described in our paper, each model produces 10,000 GPQA responses, contributing meaningfully to the 747,500 total responses in the benchmark, which is the largest calibration-focused UQ benchmark for LLMs to our knowledge.
>
> Concerning format choices, verifiable ground truth is essential for computing UQ metrics such as ECE and AUROC. Open-ended scientific QA does not provide reliably verifiable correctness labels and therefore cannot support quantitative UQ benchmarking. This constraint is standard in the literature; all six works cited by the reviewer also rely on MCQA and arithmetic QA. We address the limitations of MCQA by using APriCoT prompting to emulate open-question behaviour and complementing it with open-ended arithmetic QA, which offers fully verifiable answers and helps stabilize and validate the observed calibration patterns. Taken together, unlike many existing benchmarks that rely on synthetic or narrow tasks, our dataset selection captures challenges that are representative of real-world, high-stakes scientific reasoning.
>
>
> We thank the reviewer for their constructive input and will consider the highlighted points in the revised manuscript. We hope that our responses and the forthcoming revisions will address the concerns raised and provide a clearer basis for positively evaluating the contribution of our work.

---

### Official Review · Reviewer_eBaw · 2025-10-31

**Soundness:** 3
**Presentation:** 2
**Contribution:** 1
**Rating:** 2
**Confidence:** 4

**Summary:**

The authors construct a benchmark of science QA datasets, and evaluate several uncertainty quantification methods with large models. They analyze confidence for various types of models (such as reasoning or instruction-tuned models) and find polarization in token probabilities in reasoning and instruction-tuned models.

**Strengths:**

1. I have not previously seen an analysis of polarization for reasoning models separately from instruction-tuned models.
2. The paper is well-written and organized.

**Weaknesses:**

1. This work’s main weakness is its lack of novelty. Fundamentally, the benchmark consists of already-published datasets, there are no new methods examined, and most of the conclusions drawn by the authors have already been mentioned in previous literature—for instance, as they point out, there have been previous papers which find that instruction-tuned models are poorly calibrated.
2. I don’t agree with the assertion that “The standard evaluation technique for UQ methods are calibration plots”—there are many methods of evaluating UQ methods, including calibration plots, but also including ECE, AUROC, PRR, etc. Calibration plots additionally have several limitations—binning strategies, number of bins, etc all can majorly affect the way they look, and their interpretation is inherently qualitative which can be difficult to compare with many different configurations. This can be seen in figure A.1, where in order to fit all configurations on the screen the text and labels are illegibly tiny. With this in mind, I also disagree with listing use of summary statistics as a limitation of previous benchmarks, as these are widely used in the field for evaluation.
3. This benchmark examines many fewer UQ methods than previous work, and many are discarded because they produce unnormalized scores. In these, metrics such as AUROC could still be used for evaluation.
4. One limitation the paper claims to be addressing is narrowness of domain, but scientific QA is arguably equally narrow to most of the factual QA datasets that these UQ methods have been evaluated on already.

**Questions:**

I’m curious how increasing temperature would change the distribution for the reasoning models—is there any setting at which the polarization begins to approach that of instruction-tuned models, or does increasing temperature degenerate performance too fast?

---

> ### Author Response · Authors · 2025-11-20
> **W1 Lack of Novelty**
>
> Thank you for your thoughtful review and for engaging with our work. We address the concern below and will reflect these clarifications in the revised manuscript.
>
> While methodological novelty is important, comprehensive and realistic benchmarks are equally crucial for advancing uncertainty quantification (UQ) in LLMs. Many recent UQ methods rely on a small set of core metrics that have not been systematically evaluated in unconstrained question answering. The field therefore lacks a broad empirical basis for assessing the validity of existing or emerging approaches.
>
> Our benchmark directly addresses this gap. It includes 747,500 responses from 13 models across five providers, evaluated on seven diverse datasets spanning fact retrieval, symbolic reasoning, and open-form arithmetic QA. To our knowledge, this constitutes one of the most extensive evaluations of core UQ methods for LLMs and counters the current reliance on narrow or highly simplified validation settings.
>
> We believe that high-level observations have appeared before, but our study provides several new insights: We identify structural patterns in token-level confidence distributions, including systematic shifts and polarization (see section 6.2, p. 6), that extend beyond prior reports of poor calibration. To our knowledge, prior evidence of such polarization is limited to a single study in a narrow, binary-classification setting, and our manuscript’s phrasing inadvertently suggested a broader literature. We will revise the text to clarify this point. We further show that widely used metrics such as ECE can appear overly optimistic when high accuracy coincides with strong polarization towards high confidence scores (see section 6.3). This masking effect has not been documented at comparable scale and has direct implications for many recent UQ studies that rely solely on ECE computed on saturated benchmarks. See for example, Kapoor et al, “Large Language Models Must Be Taught to Know What They Don’t Know”, 2024, [doi](https://doi.org/10.52202/079017-2729), which is directly affected by this phenomenon.
>
> We also contribute full transparency by releasing all code, prompts, configurations, and raw uncertainty scores. Such openness is uncommon in prior work, including influential technical reports (e.g. [arxiv](https://arxiv.org/abs/2303.08774)), and we consider a rigorous, reproducible benchmark an essential and novel contribution to the field. Related to this, we note that according to the ICLR reviewer guide, novelty is not the sole criterion for evaluating submissions: “Different objectives will require different considerations as to potential value and impact \[…\] b. Strong points: is the submission clear, technically correct, experimentally rigorous, reproducible \[…\] Weak points: is it weak in any of the aspects listed in b.?” Our work aims to provide precisely the rigorous and reproducible empirical foundation that the field currently lacks from our point of view.

---

> > ### Comment · Reviewer_eBaw · 2025-11-24
> >
> > Thank you for your thoughtful response. I do not feel that most of my concerns have been addressed:
> > W1: I disagree that patterns in polarization have not been documented before: see, for instance, "Calibrated Language Models and How to Find Them with Label Smoothing" by Huang et al. or "On the Calibration of Large Language Models and Alignment" by Zhu et al. I similarly disagree that the confounding effect of accuracy on ECE has not been studied: for instance, see "Re-Examining Calibration: The Case of Question Answering" by Si et al. While I acknowledge that novelty should not be the single determining factor of acceptance, I feel that the potential value and impact of this paper is low, as it consists of datasets which were already publicly available and models which are similarly publicly available, and does not seem to demonstrate significantly new insights. Additionally, I feel that running these experiments with LM-Polygraph (which has high reproducibility) would provide significantly more metrics and UQ methods than the methods selected here.
> >
> > W2: Thanks for clarifying the difference in these two tasks. I feel that the focus specifically on a subtask of UQ (calibration) reinforces my belief that this paper would be low-impact, as there are other benchmarks such as LM-polygraph which study both calibration and discriminative power. It would be straightforward to make calibration plots for benchmarks which currently use summary statistics, as (for instance) sklearn has a straightforward implementation of a calibration display which can be constructed from confidences and labels.
> >
> > W3: See above-- the choice to ignore aspects of UQ beyond calibration feels like a major limitation in the scope of the benchmark. I would also note that this framing means the title suggests a broader scope than is examined in the paper; consider referencing calibration more explicitly, rather than uncertainty estimation.
> >
> > W4: I appreciate the point that many scientific QA datasets are indeed saturated, and would agree that "narrow" is not an accurate description of the set of tasks. That being said, the fact that it is scientific QA (rather than math, common sense, etc) feels entirely irrelevant to the paper aside from motivation (that it is important to be accurate for scientific QA), and accuracy/hallucination mitigation are generally useful for many tasks. Without any specific changes to uncertainty quantification or in-depth analysis on how UQ differs for scientific QA, this feels like a choice that makes no impact, where a different set of datasets could be substituted with no difference. It may be worth running similar UQ experiments with non scientific datasets to compare these results; I would be very interested to see if models operate differently on MCQA in scientific domains vs other domains, and believe this would strengthen the paper.
> >
> > Q1: As temperature scaling is a well-established calibration method, I would be curious to see how effectively temperature scaling could be applied to the model's distribution at inference time for token-level methods, as this feels like a method of mitigating the token polarization you see here. I have not previously seen an analysis of how post-hoc calibration methods are impacted by instruction tuning.
> >
> > I am happy to reply further if there are new points to discuss, but as it stands I feel that this work would be low impact and maintain my current score.

---

> ### Author Response · Authors · 2025-11-20
> **W2 Standard Evaluation Technique for UQ Methods in Calibration**
>
> Thank you for raising these points. We agree that clearer terminology is needed. UQ in LLMs involves two separate evaluation tasks: calibration and selective prediction (Liu et al, 2025, [arxiv](https://arxiv.org/abs/2503.15850)). Calibration assesses whether normalized uncertainty scores correspond to true probabilities of correctness, while selective prediction evaluates how well scores separate correct from incorrect answers.
>
> Within the calibration task, the standard tools in the UQ-for-LLMs literature are calibration plots and their associated summary statistics, most notably ECE. Metrics such as AUROC and PRR are indeed widely used, but they evaluate selective prediction rather than calibration. Since our benchmark focuses on calibration, the relevant comparison is to prior calibration-focused work, where calibration curves and summary measures derived from them are the primary evaluation instruments.
>
> ECE is not an independent evaluation metric. It is a summary statistic of a binned calibration plot and therefore inherits the binning sensitivities the reviewer mentions. It also compresses the entire uncertainty-score distribution into a single number, which can obscure systematic structure. In our study, we show multiple cases where models are severely miscalibrated despite reporting low ECE values, particularly when high accuracy coincides with strong overconfidence. This “masking” effect is common on saturated datasets and can make models appear well calibrated even when the curves indicate the opposite.
>
> This issue is nontrivial because many recent UQ papers rely exclusively on ECE computed on high-accuracy benchmarks, which can lead to misleading conclusions. When summary statistics are used without inspecting calibration curves or the underlying score distributions, they can hide severe miscalibration and thereby distort evaluations of new UQ methods that depend on token- or sequence-level confidence behavior. These limitations propagate through the literature as new methods build on earlier ones, potentially steering research based on inaccurate assessments of reliability.
>
> We acknowledge the concern regarding figure readability. Nonetheless, the qualitative information conveyed by calibration plots is indispensable for revealing structural patterns such as systematic overconfidence, polarization, and other miscalibration effects. These properties are clearly visible in the curves but are entirely absent from summary metrics like ECE, which is why we consider calibration plots essential in our evaluation.

---

> ### Author Response · Authors · 2025-11-20
> **W3 Amount of UQ methods and scope of the Benchmark**
>
> We appreciate the reviewer’s comment and would like to clarify the scope of our benchmark. As noted earlier, UQ in LLMs encompasses two primary tasks: calibration and selective prediction (see Liu et al, 2025, [arxiv](https://arxiv.org/abs/2503.15850)). Calibration inherently requires normalized uncertainty scores, since metrics such as ECE and calibration plots rely on probabilistic interpretations. Selective prediction, while unquestionably useful and widely applied, is a different task with different requirements and does not rely on normalized scores.
>
> Our benchmark is deliberately focused on calibration, one of the two central objectives of UQ, and therefore necessarily emphasizes methods that produce normalized uncertainty estimates. This choice is not a limitation, but rather a principled definition of the benchmark’s coverage (see first two lines in section 2).
>
> For completeness and to support broader analysis, we still report AUROC values (see various figures in the appendix) and provide the full set of raw uncertainty scores in our public repository. This allows researchers to extend our evaluation to selective prediction settings or to include methods that rely on unnormalized signals. Our goal is to provide a rigorous and reproducible foundation for calibration-focused UQ evaluation, while enabling the community to build upon our results for additional UQ tasks.

---

> ### Author Response · Authors · 2025-11-20
> **W4 Scientific QA and Domain Narrowness**
>
> Thank you for the comment. We respectfully disagree that scientific QA is “equally narrow” to the factual QA datasets commonly used for UQ evaluation. Our work aims to assess calibration quality and demonstrate that relying solely on ECE is insufficient. Treating scientific QA as equivalent to standard factual QA overlooks two key distinctions.
>
> First, most factual QA datasets used in prior UQ work, such as MMLU, are heavily saturated (see MMLU-redux, MMLU-Pro, MMLU-Pro+ etc): modern models achieve very high accuracy and tend to be highly confident. This combination triggers the central failure mode of ECE discussed in Section 6.3, where severe miscalibration can be entirely masked by high accuracy. This limitation affects many existing studies that rely exclusively on ECE. In contrast, our benchmark intentionally includes unsaturated datasets such as GPQA and SciBench, which require symbolic and multi-step reasoning rather than factual recall. As noted in Section 3.1, these tasks introduce additional challenges for UQ and expose failure patterns that saturated factual datasets cannot reveal.
>
> To address these shortcomings, we include reasoning-heavy datasets and provide full calibration plots and confidence distributions (Figure 1 and Appendix A.4.4), rather than relying on single-number ECE statistics. This yields a more informative assessment of calibration behavior.
>
> Second, the four methods we benchmark represent foundational UQ approaches from which many derivative techniques are constructed. Evaluating these methods across datasets with varying difficulty, structure, and reasoning requirements is essential for understanding their reliability in realistic use cases. Our benchmark spans fact retrieval through multi-step scientific reasoning and uses prompting strategies designed to approximate practical LLM usage while retaining verifiable ground truth.
>
> Our empirical results (Figure 2) show that two of the core UQ methods suffer substantial calibration failures and extremely low correlation between uncertainty and correctness across multiple datasets and models. These findings highlight limitations of prior literature that relied on ECE on saturated datasets, potentially producing overly optimistic conclusions.
>
> For these reasons, our benchmark fills a genuine gap by providing a comprehensive and reproducible evaluation of core UQ methods in settings that go well beyond narrow factual recall. This aligns with the ICLR’26 reviewer guide, which emphasizes the importance of work that “Is to \[...\] address a known application or problem or draw\[s\] attention to a new application or problem.”

---

> ### Author Response · Authors · 2025-11-20
> **Q1 Effect of Temperature**
>
> To provide a precise and meaningful answer, we would appreciate clarification regarding which stage of our evaluation pipeline the question refers to. Temperature can influence model behaviour at several points: (a) during the label-probability experiments, (b) during response generation for the sequence-level experiments, or (c) during follow-up queries required by methods such as Verbalized Uncertainty or P(True). Could you please specify which setting you have in mind? We would be glad to address the question in detail if time permits once this is clarified.
>
> We thank the reviewer for their constructive input and will consider the highlighted points in the revised manuscript. We hope that our responses and the forthcoming revisions will address the concerns raised and provide a clearer basis for positively evaluating the contribution of our work.

---

> ### Author Response · Authors · 2025-11-26
>
> Thank you for taking the time to respond and for your openness to further discussion. I welcome the chance to clarify our contributions and address the points you raised.
>
> **Polarization is not the same as overconfidence** Overconfidence (see Huang et al. Figure 1), describes the systematic tendency of confidence scores to exceed accuracy and does not make any claims about the shape of the predictive distribution. The polarization we describe is a distributional collapse in token-level probabilities, where the model assigns nearly all probability mass to a single label while assigning probabilities close to zero to all alternatives. This is a different phenomenon.
>
> **“Calibrated Language Models and How to Find Them with Label Smoothing” by Huang** We were unable to find any reference to distributional shifts or polarization in this paper. The authors measure calibration degradation after instruction tuning using calibration error, which does not provide insight into the distributional behaviour we characterize. If we are missing a specific statement, we would appreciate an exact pointer. We also note in line with the **Reviewer Guide**, that the cited paper was released on 01.08.2025, therefore falls outside the required comparison window and should not affect the review.
>
> **“On the Calibration of Large Language Models and Alignment” by Zhu et al. (2023)** This study analyses instruction tuning by training models of size 70M to 12B in a controlled setup. This setting is not reflective of modern open-weight LLMs or their large-scale training pipelines. The effect shown in Appendix B and Figure 7 is also fundamentally different from the phenomenon we report. The authors show concentration in an interval of width approximately 0.2 at mid-range probabilities for larger models on their own custom pipeline. We show a near degeneracy of the distribution, where single tokens reach probability close to 1.0 and all others approach 0.
>
> **To our knowledge, the only prior work explicitly reporting a similar polarization behaviour is [Cruz et al. (2407.14614)](https://arxiv.org/abs/2407.14614) and this is limited to binary classification on a specific dataset.** Our contribution is to show that this behaviour is systematic across a broad range of tasks, from fact retrieval to multi-step scientific reasoning, and across a wide range of model sizes, architectures, and variants. Our results show systematic differences across providers, implicating training pipelines in the emergence of polarization. This identifies a valuable research direction for the community, on how training pipelines can be built to mitigate this polarization effect.
>
> **We are aware that the limitations of ECE have been reported previously.** We do not claim novelty on that point. What is new is that, under the behavioural patterns we uncover (extreme token-level polarization and heavily biased verbalized confidence distributions) these limitations become critical. Since many UQ methods use token-level or verbalized signals, the behaviour we observe consistently triggers ECE’s failure mode, calling into question the robustness of earlier work that relied exclusively on ECE.
>
> **Benchmarks do not need to introduce new models or datasets.** Our contribution is to evaluate a diverse range of existing models and datasets in a setting that has not been explored in this combination previously. We present new empirical findings, including systematic polarization induced by instruction tuning, severe miscalibration in verbalized approaches, strong biases in score distributions of verbalized uncertainty and consistent method-specific patterns across a diverse set of models and datasets. These findings concern core UQ methods that serve as the foundation for many derivative approaches. Based on over a year of work on this project and an extensive survey of the literature, we are confident that this contribution provides meaningful value to the community.
>
> We initially used **LM-Polygraph** but developed our own framework because LM-Polygraph faced efficiency limitations, critical for generating the 747,500 long-form responses included in this benchmark. Our framework preserves high reproducibility, as detailed in the Reproducibility Statement. The scientific contribution of the paper is the empirical analysis, not the infrastructure used to run it, and therefore the choice of framework should not influence the evaluation of the work.
>
> **We relied on the LM-Polygraph survey for method inclusion and exclusion** and provided justification in Appendix A.4\. We are extending this section in the revision. Rejecting the benchmark based solely on the number of included methods, without regard to their applicability, computational feasibility, and relevance in the field, risks overlooking the qualitative criteria that determine whether a method is suitable for large-scale long-form scientific QA.

---

> ### Author Response · Authors · 2025-11-26
>
> **Every paper must define a scope.** LM-Polygraph focuses on many metrics but evaluates them only on three models and one QA dataset with short-form answers, only assessing summary statistics. We instead focus on calibration in realistic settings involving long-form responses requiring multi-step reasoning across a diverse range of model providers, variants, architectures and sizes and datasets spanning 2 tasks and a range of difficulty levels. Our priority is not the sheer number of UQ methods but the relevance and interpretability of the selected core methods in realistic long-form scenarios and depth of analysis. We therefore disagree that focusing on one core task constitutes a major limitation. Scope is determined by depth and suitability for the target domain, not raw quantity.
>
> We state clearly that scientific QA induces distinct challenges: long responses, extended reasoning chains with interdependent steps, and syntactic complexity such as formulas and MathML. These characteristics are central to the evaluation setting. **Our datasets were selected to ensure these challenges appear in the generated responses.** Although hallucination mitigation is broadly relevant, it is particularly important for complex reasoning tasks where verification is difficult. By including datasets such as MMLU and GSM8K, which induce far fewer of these challenges, we provide a direct comparison. We observe that the methods behave systematically across both low- and high-complexity datasets, which supports the generalizability of our findings. We will emphasize this more clearly in the revision.
>
> **Concerning Q1** In the label-probability experiments we report a near-complete collapse of probability mass onto a single token. The phenomenon involves extremely large logit margins and therefore reflects a distributional collapse rather than a mild calibration error. In such cases, temperature scaling is ineffective as it cannot meaningfully adjust such distributions, because the differences between logits are dominated by noise once the margin is this large and no longer contain calibration-relevant information.
>
> I hope these clarifications help address the concerns raised, and I appreciate your continued consideration of our work.

---

### Official Review · Reviewer_KGe9 · 2025-11-12

**Soundness:** 2
**Presentation:** 3
**Contribution:** 2
**Rating:** 4
**Confidence:** 4

**Summary:**

The paper introduces an open-source benchmark to evaluate UQ for LLMs on QA tasks, spanning 11 base, instruction-tuned, and reasoning models across 8 datasets. It finds that instruction tuning polarizes token probabilities, making token-level confidences poorly calibrated, while at the sequence level verbalized self-reported confidence is biased and weakly correlated with correctness. In contrast, self consistency across sampled generations is the more reliable but computationally expensive signal.

**Strengths:**

- Establishes a large-scale benchmark focused specifically on uncertainty estimation for QA, spanning 11 LLM variants and 8 datasets.
- Provides clear empirical insights: instruction tuning polarizes token probabilities and consistency via answer frequency is a stronger sequence-level signal.
- Covers diverse difficulty levels and domains to stress-test UQ under both factual recall and multi-step reasoning.
- Open-sources a repo for reproducibility.

**Weaknesses:**

- I personally find the focus on multi-choice QA unreasonable and unrealistic. Most of the time in practice LLMs are hardly applied to some questions with a pre-defined answer set. Therefore evaluating the model uncertainties under this limited scenario does not reflect their behavior on open-ended QA and bears minimal practical impact.

- **Most of the observations, unfortunately, are defined and discussed (and addressed to some extend) more rigorously in a not-so-recent paper [1]**. The previous work also addressed the accuracy issue while calculating ECE by balancing the correct/incorrect answers. The contribution and novelty of this paper is greatly undermined given this previous work.

- Missing related works, e.g., [2].

- This paper focuses on scientific QA, but none of the discussion is specific the the special characteristics of the scientific domain (e.g., how the questions and answers differ from the general domain?)

[1] Li Y, Qiang R, Moukheiber L, Zhang C. Language model uncertainty quantification with attention chain. arXiv preprint arXiv:2503.19168. 2025 Mar 24.
[2] Wang X, Zhang Z, Chen G, Li Q, Luo B, Han Z, Wang H, Li Z, Gao H, Hu M. Ubench: Benchmarking uncertainty in large language models with multiple choice questions. InFindings of the Association for Computational Linguistics: ACL 2025 2025 Jul (pp. 8076-8107).

**Questions:**

What findings are specific to the scientific domain?

---

> ### Author Response · Authors · 2025-11-20
> **W1 Choice of Datasets**
>
> Thank you for your thoughtful review and for engaging with our work. We address the concern below and will reflect these clarifications in the revised manuscript.
>
> We respectfully disagree that the usage of MCQA is “unreasonable” or lacks practical impact. UQ fundamentally depends on ground-truth, so verifiable tasks to compute calibration and selective prediction metrics such as ECE and AUROC. MCQA has therefore been a central and established evaluation setting in UQ for LLMs. This is reflected broadly in the literature, including the GPT-4 Technical Report, novel method validations, benchmarks, and the works cited by the reviewer: \[2\] introduces a benchmark composed exclusively of MCQA, and \[1\] evaluates on three datasets including BigBenchHard, which is multiple-choice. Moreover, BigBenchHard includes tasks such as “penguins in a table” and “geometric shape parsing” from SVG input, which are substantially less representative of practical LLM use than the MCQA datasets we employ.
>
> We acknowledge that MCQA has limitations as a proxy for open-ended QA. Our benchmark addresses this in two ways. (a) Following \[1\], we complement five MCQA datasets with three open-ended arithmetic QA datasets, which require free-form answers. Importantly, our main findings hold across both MCQA and open-ended settings, supporting their generality.
>  (b) To mitigate the restrictiveness of MCQA formats, we adopt the APriCoT prompting strategy, which elicits explicit reasoning about each candidate and therefore better approximates open-ended QA behaviour. Reasoning over incorrect options induces hallucination-prone settings, which aligns with the failures UQ methods aim to detect.
>
> Our dataset selection is deliberate and designed to provide a comprehensive assessment of UQ for natural-science question answering, covering factual retrieval, symbolic reasoning, and multi-step analytical reasoning. This avoids the overly narrow evaluation settings of prior work, such as CCP’s constrained biography reproduction tasks or the limited scope of \[1\], which may overestimate method robustness. In contrast, our benchmark reveals substantial gaps between controlled validation of previous papers and realistic performance. Given the rapid emergence of derivative methods built on core approaches that we evaluate and in some cases find uncorrelated with accuracy, our benchmark provides essential empirical grounding for the field.

---

> ### Author Response · Authors · 2025-11-20
> **W2 Relation to prior work [1]**
>
> We thank the reviewer for raising \[1\]. While there is partial overlap, the claim that our contribution is “greatly undermined” does not reflect the scope or novelty of our work. Our benchmark evaluates 747,500 responses across eight datasets and eleven contemporary LLMs, including instruction-tuned and reasoning-specialized models from five providers ranging from 7B to 70B parameters. We assess four widely used foundational UQ methods that have inspired many derivative approaches. The contribution of our work is not a new method but a systematic, large-scale evaluation of these techniques on realistic QA tasks, rather than narrowly constrained settings common in prior work, including those used in \[1\].
>
> We provide full transparency through public release of code, configurations, prompting templates, raw uncertainty scores, and a reproducible benchmarking framework to support robust evaluation of UQ methods.
>
> While \[1\] focuses primarily on evaluating its proposed techniques, our study extends their scope in several important ways. We additionally evaluate token-level probability calibration using both label probability and P(True) as sequence-level metrics, which are central for classifier-style use cases and probability-based UQ methods. We also conduct a broader and deeper analysis of the four core approaches. Beyond calibration errors, we analyse its certainty score distributions, showing that models are providing polarized scores, constraining themselves to a narrow range and that correlations between uncertainty and correctness are extremely low across nearly all models for methods that were previously “proven” as effective. This discrepancy persists even when ECE appears low, demonstrating that ECE can be misleading without inspecting the underlying score distribution. We show that high accuracy combined with polarization of scores can mask severe miscalibration, a concern given that many recent papers rely solely on ECE computed on oversaturated datasets, risking misleading baselines and propagated errors.
>
> Although \[1\] notes the dependence between accuracy and ECE, its proposed mitigation via “balancing the correct and incorrect answers” appears only in the appendix and raises methodological concerns. Forcing a 50:50 split effectively imposes 50 percent accuracy, alters the prediction distribution, and does not remove ECE’s intrinsic dependence on accuracy. The instability of their approach, which is acknowledged by the authors using five different balanced splits and averaging, further illustrates its lack of robustness. Addressing this issue is central to our conclusions, essential for reliable UQ evaluation and a valuable finding for reliable evaluation of existing and future methods.

---

> ### Author Response · Authors · 2025-11-20
> **W3 Missing related work [2]**
>
> We appreciate the reviewer’s comment and acknowledge the relevance of \[2\]. We began this project in October 2024 and conducted a comprehensive literature survey at that time, which included UBENCH in its initial June 2024 release. While \[2\] presents a promising multi-faceted MCQA dataset, we were unable to include it because the data was not publicly available. The dataset remained closed source during our preparation, and although the updated paper claims an expansion to 12,000 items, the [public repository is still largely empty, with the README indicating “Quick Start: todo”](https://github.com/Cyno2232/UBENCH). Without access to the dataset, benchmarking or reproducing results is not possible.
>
> We reviewed nearly all relevant work published up to mid-2025 and continued monitoring developments thereafter. As with any submission, we cannot cite every paper; our citation criteria require community accessibility, review status, and direct methodological or empirical relevance. While \[2\] may become an important contribution, we cannot integrate it until its core dataset is released to the public research community.

---

> ### Author Response · Authors · 2025-11-20
> **W4 Focus on Scientific QA**
>
> We respectfully disagree with the claim that our paper does not address characteristics specific to scientific QA. In Section 3.1, we explicitly discuss domain-specific challenges, including higher factual density, specialized vocabulary, multi-step reasoning requirements, and the increased risk of plausible but incorrect answers.
>
> Our dataset selection and prompting strategies were chosen to reflect realistic scientific use cases, in contrast to prior benchmarks that either operate in overly narrow domains or rely on highly artificial tasks such as those in BigBenchHard from \[1\], which do not represent practical scientific applications.
>
> Our benchmark evaluates 747,500 responses across four UQ methods, providing a large-scale quantitative assessment of uncertainty behaviour in scientific QA. A full qualitative analysis of responses is infeasible at this scale. Instead, we examine UQ applicability to scientific contexts by evaluating models on datasets that directly incorporate the domain-specific challenges outlined in our discussion.
>
> We thank the reviewer for their constructive input and will consider the highlighted points in the revised manuscript. We hope that our responses and the forthcoming revisions will address the concerns raised and provide a clearer basis for positively evaluating the contribution of our work.

---

> ### Comment · Reviewer_KGe9 · 2025-11-22
>
> Thanks for your responses.
>
> For W1, I did not say you cannot use MCQA; but totally depending on it would be concerning. I appreciate your adding more tasks.
>
> For W2, the problem is that [1] has answered RQ1, and covered about a half of the content of RQ2. In fact, if you look at the abstract lines 021--027, which I guess is the main selling point of this work, all of these observations are already a part of discussion covered by the previous work. I agree that the previous work is not as comprehensive in terms of baselines and datasets but such overlap could still impact the contribution.

---

> > ### Author Response · Authors · 2025-11-23
> > **Differentiating Our Contributions from Prior Work [1]**
> >
> > **Thank you for your follow-up!** We believe there is still a fundamental misunderstanding regarding the contributions of our work compared to \[1\] and we will clarify these points more explicitly in the revised manuscript. Our contributions comprise answering (1) RQ1 and (2) RQ2 and (3) highlighting and contextualizing a critical limitation of ECE that emerges directly from the phenomena we study.
> >
> > **(1) RQ1 - Calibration of Token-Level Probabilities**
> >
> > The work in \[1\] does **_not_**address RQ1\. Although \[1\] _uses_ token-level probabilities for the subset of tokens identified as important via its novel attention backtracking approach, it does not study whether these probabilities are _calibrated_, nor does it investigate their distributional behaviour. In contrast, our work explicitly evaluates whether token-level probabilities constitute a reliable measure of uncertainty for LLMs. We build on two prior foundations: the [GPT-4 Technical Report](https://arxiv.org/abs/2303.08774), which first raised the question of token-level calibration in instruction tuned models and [Cruz et al. (2024)](https://arxiv.org/abs/2407.14614), who observed limited polarization in a binary setting for Yes/No answers for instruction tuned models. Our study generalizes and substantially extends this assessment of token-level calibration.
> >
> > Specifically, we evaluate a broad set of models across providers, sizes, architectures and types (including reasoning models) on diverse difficulties (fact retrieving, symbolic reasoning, multistep reasoning). We uncover model-family specific differences (e.g. Mistral models), suggesting that polarization likely arises from training-pipeline choices. To the best of our knowledge, \[1\] does not investigate or discuss calibration, polarization, instruction tuning or reasoning models _in the context of token-level probability distributions_.
> >
> > **(2) RQ2 - Calibration of Core Sequence-Level UQ Methods**
> >
> > For RQ2, \[1\] evaluates only Verbalized Uncertainty on a comparatively narrow domain. Their work does not study P(True), CCP or Frequency of Answer. Their “semantic-consistency” aggregation simply averages scores across 5 samples, is therefore not an atomic evaluation of semantic consistency and cannot assess the actual contribution of semantic consensus as a UQ signal.
> >
> > Our work provides depth and breadth: we analyze core UQ methods in detail (e.g. distributional bias of Verbalized Uncertainty) and benchmark them across a wider range of models and datasets. This allows us to surface cross-model contrasts that are essential for understanding where and why methods fail. This information is also absent from \[1\].
> >
> > Moreover, our focus on core UQ approaches is deliberate: many emerging methods, including those in \[1\], depend directly on token-level scores or derive components from fundamental UQ methods we research. By analyzing these foundations rigorously, our results are broadly relevant to to the field.
> >
> > **(3) Limitations of ECE**
> >
> > While we acknowledge that \[1\] also notes limitations of ECE, our contribution is to situate this issue directly within the empirical phenomena we uncover. We show that ECE fails specifically under extreme polarization (e.g. token-level probability distributions or CCP) and highly biased uncertainty distributions (e.g. Verbalized uncertainty). Our results demonstrate that UQ methods for LLMs routinely induce this failure mode, meaning that conclusions drawn from ECE-based evaluations in prior work may be systematically compromised, characterizing this as a severe effect for the field. Elevating the observation from a conceptual remark to an empirically supported finding is (to our knowledge) new.
> >
> > **Degree of Overlap with \[1\]**
> >
> > We agree that there is partial overlap on two points: **Verbalized Uncertainty underperforms** and **ECE can become unreliable as measure for calibration**. Beyond these two high-level conclusions, **the overlap is narrow**. Our study provides large-scale validation across many more models and datasets, identifies phenomena (including detailed characterization of polarization) and contextualizes their implications for future LLM development and UQ methodology. **We therefore believe our broad and systematic analysis is of high value to the field.**
> >
> > Finally, we fully open source our benchmark code, raw uncertainty scores and all plots, along with a framework for efficient and reproducible LLM-UQ benchmarking, supporting ongoing research, including future method-centric work such as \[1\].
> >
> > We acknowledge that at first glance the papers may appear similar, but their goals, scopes and contributions differ substantially. We will revise the manuscript to articulate these distinctions more clearly. If there are specific aspects that remain unclear or would help us further improve the presentation, we would be grateful for additional guidance.
> >
> > **Thank you again for your thoughtful engagement and for helping us strengthen the paper!**

---

### Author Response · Authors · 2025-12-03
**Paper Revisions Summary**

We again thank the reviewers for their feedback and summarize below the key revisions made to strengthen the manuscript:

* Following reviewer hmfZ’s valid concern regarding invalid answers, we reran experiments using structured decoding (newly supported since the initial experiment design) and increased token limits for reasoning models. This **substantially reduced invalid outputs and reinforces the robustness of our conclusions (see Table A.5 for comparison).**
* We revised the related work and **clarified our core contributions, explicitly positioning our benchmark relative to prior studies.** We emphasize the importance of scientific QA as a testbed for UQ due to its unique reasoning demands.
* We **expanded the discussion of ECE’s limitations** to more clearly convey why these issues become critical in the presence of the observed model behaviors (polarization and biased uncertainty estimates), without presenting this as a novel discovery. (see Section 6.3)
* We **added clarifications on the scientific domains covered by the selected datasets** (see Section 3.3)
* To address recurring reviewer questions, we extended the appendix with detailed explanations of **method selection, the effects of temperature, and mitigation strategies for invalid answers**. (see Sections A.5, A.9, A.6)

We hope these revisions improve the clarity and contribution of our paper and effectively address the weaknesses and questions raised in the reviews.

---

### Meta-Review · Area_Chair_N4F7 · 2025-12-11

**Summary:**

After reviewing the paper myself and familiarizing myself with the entirety of the Reviewer-Author discussion, I conclude I agree with the entirety of feedback of Reviewer eBaw and most of the feedback of Reviewer hmfZ (W5 might be asking too much in terms of scope, and would not be necessary if other contributions of this paper were novel). In summary their review shows concerns about the novelty and methodology.

**Reviewer Concerns:**

While the authors addressed some of the addressable feedback in the course of rebuttal, I share the concerns about the novelty of the work and the significance of the novel results, as those cannot be easily addressed.

I find the most novel contribution of this paper to be the evaluation of reasoning models in comparison to their non-reasoning counterparts, e.g. Figure 1. I think a similarly careful evaluation and comparison in Section 7 would further strengthen the paper and offer opportunities for analysis of the results that maybe would allow creating insights novel enough to warrant publication at a venue like ICLR.

As it stands, the novelty of these results is too small to warrant publication. Benchmarking papers are worthy of acceptance in a venue like ICLR if they are particularly novel in terms of insights, or offer evidence countering conventional knowledge/understanding in the field, or make contribution in terms of particularly careful evaluation methodology. Unfortunately, I do not find this paper to meet the bar on these criteria.

I thank the authors for their changes in the latest paper revision. They addressed the main factual correctness issues I previously recognized in this work, in my opinion, making it close to ready for acceptance in a venue like TMLR.

I remain dubious about the focus on “scientific QA”, but it is not a key factor in my decision-making. The empirical evidence does not explicitly compare how the performance of UQ methods compare in these settings with “unique reasoning demands” vs. settings the authors consider different. Like some of the reviewers, I doubt that we would observe a difference.

**Reviewer Scores:**

* eBaw (2->4): While the reviewer's first replied "I do not feel that most of my concerns have been addressed", the reviewer could have raised their score to 4 if they had agreed with some of the authors' last response.
* hmfZ (2->4): while the reviewer did not reply to the authors, I believe some of their concerns were addressed in the rebuttal. However, the significance and novelty concerns would have remained most probably.
* KGe9 (4->4): while for different reasons, this reviewer would share concerns about the significance of this work with  eBaw and hmfZ, and probably would have agreed on the same score.
* nhTz (4): I will not take this review into account since it is superficial most probably LLM generated

---

### Decision · Program_Chairs · 2026-01-26

Reject